# A Linearly Convergent Proximal Subgradient Algorithm for Sparse Portfolio Optimization with Transaction Cost

**Xiaoting Yao** [1]  **Na Zhang** [2]

## Abstract

Transaction cost optimization (TCO) of online portfolio selection is crucial in computing science, due to the significant impact of transaction costs in practical short-term trading. Moreover, sparsity of portfolio vector is often desired to enhance stability and decrease risk. However, there is a lack of models considering transaction costs and sparsity simultaneously in the literature. In this paper, we first propose a $K$-sparse TCO model that minimizes the negative return and transaction costs while keeping the portfolio vector being $K$-sparse. Noting that the model is NP-hard due to the $K$-sparse constraint, we bypass this difficulty by reformulating the sparse model to a nonsmooth difference of convex (DC) optimization problem. We show that both problems are equivalent by proving that the penalty parameter is large enough. Then, to overcome the difficulty caused by the nonsmoothness and the simplex constraint of the model, we develop a proximal subgradient algorithm (PSGA) to solve the DC problem and apply the alternating direction of multipliers (ADMM) to compute the proximity operator of the corresponding function. Furthermore, we establish the global convergence of the entire sequence generated by PSGA through showing the surrogate function satisfies the Kurdyka-Łojasiewicz (KL) property. In addition, by showing the KL exponent of the surrogate function is $1/2$, we establish the R-linear convergence rate of PSGA for any arbitrary initial point. Finally, we compare our proposed algorithm with other state-of-the-art strategies on four benchmark real-market data sets, with the numerical results showing that the proposed al-

gorithm achieves lower risk while keeping higher return than classical TCO models.

## 1. Introduction

Short-term portfolio optimization (SPO) aims to seek higher returns through frequent trading, such as hourly and daily, which leads to significant transaction costs. However, most existing strategies (Lai et al., 2018b; Luo et al., 2020; Lin et al., 2024a; Li et al., 2026) do not take transaction cost into consideration, leading to poor performance in practice. Thus, transaction cost optimization (TCO) turns into a vitally important problem in SPO.

The original extension of TCO stems from (Bauer & Kohavi, 1999)'s proof that incorporating transaction costs in the Universal Portfolio framework (Cover, 1991) remains theoretically valid. The other widely adopted extension was proposed by (Györfi & Vajda, 2008), they assumed that incurred transaction cost is proportional to the wealth transferred during rebalancing. However, they either did not consider transaction cost into decision process, or could only handle simple linear conditions. In order to address complex short-term scenarios, heuristic algorithms or machine learning methods (Lai et al., 2018a; Lim et al., 2022; Zhao et al., 2021b; Corsaro et al., 2022) need to be considered. (Konno et al., 2005) proposed a branch-and-bound method to attack long-short portfolio optimization problems involving concave functions and DC transaction costs. (Das et al., 2013) considered the challenge of transaction costs by leveraging machine learning methodologies and employed a genetic programming algorithm to derive an effective solution. Further, (Li et al., 2017) were the first to exploit mathematical method for online portfolio selection with transaction costs. They clarified the relationship between net wealth and transaction costs, and further derived explicit solutions for portfolio updates. Recently, approaches based on (Li et al., 2017) that regularize transaction costs have attracted increasing attention (Pham Dinh et al., 2016; Zhou et al., 2022; Yao & Zhang, 2023b). Although these methods performed better in real investment than most previous strategies, they may suffer from poor generality due to the absence of regularization for portfolio variable.

[1]School of Computer Science and Engineering, Sun Yat-sen University, Guangzhou 510275, China [2]Department of Applied Mathematics College Mathematics and Informatics, South China Agricultural University, Guangzhou 510642, China. Correspondence to: Na Zhang <nazhang2014@scau.edu.cn >.

*Proceedings of the 43rd International Conference on Machine Learning*, Seoul, South Korea. PMLR 306, 2026. Copyright 2026 by the author(s).

On the other hand, one may require a $K$-sparse portfolio to improve the stability of the TCO model, where the number of selected assets is restricted to a $K$-size to maintain simplicity and save time and financial costs (Brodie et al., 2009; Lin et al., 2024a). Common approaches to enforcing sparse constraints include combinatorial methods, such as mixed-integer programming (Günlük & Linderoth, 2010; Bertsimas & Cory-Wright, 2022), however, research (Lin et al., 2024b) pointed out that these methods require visiting about 1300-77000 nodes, remaining higher computational burden for practical investment. Besides, reinforcement learning (RL) has also been applied to solve sparse constraints (Jiang et al., 2017; Betancourt & Chen, 2021; Wu & Li, 2024; Song et al., 2025), but as (Lai & Yang, 2022) pointed out, RL is more suitable for long-term investment, such as monthly or yearly, because its parameters need a stable distribution, and it operates as a black-box model, whereas our work prioritizes interpretability. Hence, a suitable method for the sparse portfolio problem in the short-term is based on machine learning. (Lai et al., 2018b) were the first to consider sparsity in short-term investments and proposed the short-term sparse portfolio optimization (SSPO) framework. They applied $\ell_1$ norm to portfolio variable to induce sparsity and utilized the alternating direction method of multipliers (ADMM) to obtain the closed-form solution of the model. The $\ell_1$ norm regularization, serving as a convex relaxation of $\ell_0$ to generate sparsity, has been widely adopted in SSPO (Gao & Yang, 2019; Corsaro et al., 2022; Shen et al., 2014; Zhao et al., 2021a). However, the $\ell_1$ norm often yields insufficient sparsity that can not meet the sparsity requirements in real-world investments. Therefore, (Luo et al., 2020) optimized SSPO by replacing $\ell_1$ with $\ell_0$ and obtained three kinds of closed-form solutions. Furthermore, (Wang et al., 2023) directly used the $\ell_0$ norm to construct sparse portfolios and employed ADMM for solving the optimization problem.

Although sparse portfolio selection has been extensively investigated in the past decades, existing literature seldom takes transaction costs into consideration. In this paper, we propose a $K$-sparse TCO model, which considers transaction costs and the sparsity of the portfolio vector simultaneously. The difficulty for solving this model comes from three aspects. The first one is the $K$-sparse constraint, making the model NP-hard. The second and third difficulties are from the nonsmoothness of the objective function, and the simplex constraint, respectively. In this paper, we bypass the first difficulty through reformulating the model to a difference of convex (DC) problem and tackle the last two difficulties by proximal subgradient algorithm (PSGA) and ADMM. Our main contributions are summarized as below.

- We propose a $K$-sparse TCO model, which minimizes the sum of the negative return and the transaction cost while maintaining the portfolio vector being $K$-sparse. A key innovation of our model lies in its integrated consideration of short-term investment, sparsity, and transaction costs—a combination that, to the best of our knowledge, has been rarely explored simultaneously in the existing literature.

- We bypass the difficulty caused by the $K$-sparse constraint of the model through reformulating the $K$-sparse TCO model into a nonsmooth DC optimization problem. We show that both problems are equivalent in the sense that the DC problem is an exact penalty problem of the $K$-sparse TCO model. Specifically, both problems share the same global (local) optimal solutions provided that the penalty parameter is large enough.

- In order to solve the nonsmooth DC problem, we design a PSGA and ADMM to compute the proximity operator of the corresponding function. We establish the global convergence of the entire sequence generated by PSGA by constructing a surrogate function. It is worth noting that a central theoretical novelty of our paper is that we utilize a more streamlined auxiliary function. This strategic choice allows us to establish that its Kurdyka-Łojasiewicz (KL) exponent is 1/2, which in turn directly leads to a linear convergence rate.

The structure of the paper is outlined as follows. In Section 2, we review the related works. Section 3 is devoted to the $K$-sparse TCO model as well as its DC reformulation, and the PSGA for solving it. In Section 4, we conduct numerical experiments to evaluate the effectiveness of the proposed algorithms. Finally, we conclude the paper with a summary of the key findings in Section 5.

## 2. Related Works

In this section, we review some existing works related to portfolio selection and transaction cost optimization.

### 2.1. Sparse Short-term Portfolio Selection

Short-term portfolio optimization (SPO) aims to maximize cumulative wealth in a short investment period. Hence, the classical Kelly's Exponential Growth Rate (1956) approach indicated that the selected $m$-diamond portfolio vector $\mathbf{w}$ can be directly integrated with online learning schemes over historical returns, that is,

$$\mathbf{w}^* = \arg\max_{\mathbf{w} \in \mathbb{R}^m} \sum_{t=1}^{T} \log \mathbf{w}^\top \mathbf{x}_{[t]}, \tag{1}$$

where $\mathbf{x}_{[t]}$ denotes real relative price vector in $t$-th investment period for $t = 1, 2, \cdots, T$. Thus, $\sum_{t=1}^{T} \log \mathbf{w}^\top \mathbf{x}_{[t]}$ implies the hindsight cumulative wealth after invested for $T$

periods. However, it is obvious that (1) is a hindsight strategy, which is infeasible to implement in reality. Moreover, the lack of consideration for variable sparsity in (1) results in increased risk and cost.

Recently, (Lai et al., 2018b) are the first to propose the following sparse short-term portfolio optimization (SSPO) model with $\ell_1$ regularization:

$$\mathbf{w}_{[t+1]} = \arg\min_{\mathbf{w} \in \Delta_m} -\mathbf{w}^\top \mathbf{q}_{[t]} + \lambda ||\mathbf{w}||_1, \qquad (2)$$

where $\lambda$ is the regularization parameter, $\mathbf{q}_{[t]}$ is the predicted relative price at $t$-th period, and $\Delta_m = \{\mathbf{w} \in \mathbb{R}_+^m : \mathbf{w}^\top \mathbf{1} = 1\}$ is the $m$-dimensional simplex, restricting portfolios are self-finance and no margin and no shorting. Different from (1), (2) updates the portfolio $\mathbf{w}$ in $(t + 1)$-th period, denoted as $\mathbf{w}_{[t+1]}$, by incorporating the information available from the first $t$ periods, making it feasible for real world implementation. Its performance is subsequently assessed by comparing the cumulative wealth attained over $T$ investment periods. Besides, $||\mathbf{w}||_1$ is the convex relaxation of $\ell_0$ for inducing sparse portfolios. However, the $\ell_1$ regularization in (2) may reduce the sparsity of the primal variable $\mathbf{w}$. To this end, (Luo et al., 2020) optimized SSPO and proposed following $\ell_0$-SSPO model with closed form solutions:

$$\mathbf{w}_{[t+1]} = \arg\min_{\mathbf{w} \in \Delta_m} -\mathbf{w}^\top \mathbf{q}_{[t]} + \lambda ||\mathbf{w}||_0, \qquad (3)$$

where $||\mathbf{w}||_0$ counts the number of nonzero elements of $\mathbf{w}$. In addition to the previously discussed models, SSPO literature encompasses various methodological approaches such as moving average reversion (Li et al., 2015; 2011; 2012) and risk-minimization frameworks (Lintner, 1965; Lai et al., 2020; Shi et al., 2024). Due to the limitation of space, we omit a detailed discussion of these methods.

Although the above portfolio optimization methods may achieve good sparsity, they fail to incorporate the practical consideration of transaction costs, resulting in a wide gap between theory and practice. Quantitative evidence supporting this observation is presented in Section of Experimental Results.

### 2.2. Transaction Cost Optimization

In order to achieve higher returns, frequent transactions occur in SPO. This leads to significant transaction cost. Therefore, it is evident that transaction cost optimization (TCO) problem is necessary to be considered. Li et al. (2017) are the first to apply mathematical techniques to address the problem of SPO with transaction costs. They demonstrated that transaction costs is inversely correlated to $||\mathbf{w} - \hat{\mathbf{w}}_{[t]}||_1$, which can be derived from

$$1 = \theta_{[t-1]} + \gamma ||\hat{\mathbf{w}}_{[t-1]} - \mathbf{w}_{[t]}\theta_{[t-1]}||_1,$$

where $\gamma$ is the transaction cost rate, $\theta_{[t-1]}$ is net wealth proportion implicated transaction costs in $(t-1)$-th period. Note that $\hat{\mathbf{w}}_{[t-1]}$ denotes the asset allocation at the end of $(t-1)$-th period, i.e., $\hat{\mathbf{w}}_{[t-1]} = \frac{\mathbf{w}_{[t-1]} \cdot \mathbf{x}_{[t-1]}}{\mathbf{w}_{[t-1]}^\top \mathbf{x}_{[t-1]}}$, where $\cdot$ is the element-wise product. Therefore, they attack the transaction cost problem by presenting the following TCO framework

$$\mathbf{w}_{[t+1]} = \arg\min_{\mathbf{w} \in \Delta_m} -\mathrm{E}\{\log \mathbf{w}^\top \mathbf{q}_{[t]}\} + \lambda ||\hat{\mathbf{w}}_{[t]} - \mathbf{w}||_1. \qquad (4)$$

Besides, it is worth noting that, with the inclusion of transaction costs, the approach to computing cumulative wealth has been modified. Li et al. (2017) pointed out that cumulative wealth $S_T$ invested for $T$ periods should be computed by

$$S_T = S_0 \prod_{t=1}^{T} (\mathbf{w}_{[t]}^\top \mathbf{x}_{[t]}) \theta_{[t-1]}, \qquad (5)$$

where $S_0$ represents the normalization of initial capital and it is always set to 1.

Although the TCO model (4) achieves better cumulative wealth in real investment, it may lack of stability and generality since no regularization on $\mathbf{w}$ is considered in the model. This will result in a high risk of this model. To address this issue, we will incorporate a sparse regularization term on $\mathbf{w}$ to enhance model robustness and mitigate risk.

## 3. PSGA (Proximal Subgradient Algorithm) for Sparse Transaction Cost Optimization

In this section, we first propose a $K$-sparse transaction cost optimization problem for portfolio selection and reformulate it to a DC (difference of convex) optimization problem. The equivalence of both problems is established in the sense that they have the same global and local solution sets provided that regularization parameter is larger than a prefixed number related to the problem. Then, a proximal subgradient algorithm is developed for solving the DC reformulation.

### 3.1. $K$-sparse Transaction Cost Optimization and its DC Reformulation

In order to obtain high return, reduce transaction costs, and simultaneously maintain a smaller number of assets, we propose the following $K$-sparse TCO (STCO) model:

$$\mathbf{w}_{[t+1]} \in \arg\min_{\mathbf{w} \in \Delta_m} -\mathbf{q}_{[t]}^\top \mathbf{w} + \lambda ||\mathbf{w} - \hat{\mathbf{w}}_{[t]}||_1$$
$$\text{s.t. } ||\mathbf{w}||_0 \leq K, \qquad (6)$$

where $\lambda > 0$ is a regularization parameter, and $K$ is the number of selected assets. We can take several observations in model (6). The first term of the objective function indicates the negative return, where its minimization leads to

increased wealth in a short-term. The last term $\lambda||\mathbf{w} - \hat{\mathbf{w}}_{[t]}||_1$ is a transaction cost positive related term, contributing to the objective of reducing transaction cost. Moreover, the $K$-sparse constraint enforces cardinality control by limiting the number of selected assets no more than $K$, resulting in lower risk of the portfolio selection model.

The difficulty of tackling problem (6) comes from three aspects. The first one is the $\ell_0$ constraint, making it NP-hard. The second and third difficulties are from the non-smoothness of the $\ell_1$-norm and the simplex constraint, respectively. In the following, we bypass the first difficulty through reformulating problem (6) into a difference of convex (DC) problem. The idea is motivated by the work of (Gotoh et al., 2018). It is obvious that the $\ell_0$ constraint is equivalent to the constraint: $||\mathbf{w}||_1 - ||\mathbf{w}||_{(K)} = 0$, where $||\mathbf{w}||_{(K)}$ is the vector $K$-norm of $\mathbf{w}$, defined as $||\mathbf{w}||_{(K)} := \sum_{i=1}^{K} |\mathbf{w}_{(i)}|$ with $|\mathbf{w}_{(i)}|$ denoting the $i$-th largest element of the absolute values of $\mathbf{w}$. Consequently, problem (6) is equivalently reformulated to the following model:

$$\mathbf{w}_{[t+1]} \in \underset{\mathbf{w} \in \Delta_m}{\arg\min} \; -\mathbf{q}_{[t]}^\top \mathbf{w} + \lambda||\mathbf{w} - \hat{\mathbf{w}}_{[t]}||_1$$
$$\text{s.t. } ||\mathbf{w}||_1 - ||\mathbf{w}||_{(K)} = 0. \quad (7)$$

To attack the difficulty caused by the non-convex constraint of problem (7), a normal approach is to penalizing the constraint into the objective. This leads to the following DC transaction cost optimization problem:

$$\mathbf{w}_{[t+1]} \in \underset{\mathbf{w} \in \Delta_m}{\arg\min} \; -\mathbf{q}_{[t]}^\top \mathbf{w} + \lambda||\mathbf{w} - \hat{\mathbf{w}}_{[t]}||_1 - c||\mathbf{w}||_{(K)},$$
$$(8)$$

where $c > 0$ is the penalty parameter, the term $||\mathbf{w}||_1$ is missed since $\mathbf{w} \in \Delta_m$ implies $||\mathbf{w}||_1 = 1$. Fortunately, problem (8) has a better structure than problems (6) and (7), since it is a DC problem. In the remaining part of this subsection, we dedicate to showing that the DC formulation (8) is an exact penalty formulation of problem (6).

We begin with the existence of optimal solutions to problems (6) and (8), which can be derived from the continuity of their objective functions and the compactness of their constraints. This result is presented in the following theorem.

**Theorem 3.1.** *For any $\lambda, c > 0$ and $K \geq 1$, the optimal solution sets of problems (6) and (8) are nonempty.*

In order to show the exact penalty property of problem (8), we first present in the following proposition that the distance between $\mathbf{w}$ to the constraint set $\Omega$ of problem (6) can be bounded by $p(\mathbf{w})$, where $p(\mathbf{w}) = ||\mathbf{w}||_1 - ||\mathbf{w}||_{(K)}$ and $\Omega := \{\mathbf{w} \in \Delta_m : ||\mathbf{w}||_0 \leq K\}$. Due to the limitation of space, all the proofs in this paper are placed in the appendix. The proof of Proposition 3.2 is provided in appendix A.2 .

**Proposition 3.2.** *For all $\mathbf{w} \in \Delta_m$, there holds*

$$dist(\mathbf{w}, \Omega) \leq 2\sqrt{2}p(\mathbf{w}).$$

With the help of the above proposition, we obtain the following theorem regarding the exact penalty formulation (8) for problem (6). We denote the objective functions of (6) and (8) by $f$ and $f_c$, respectively. Specifically, $f(\mathbf{w}) = -\mathbf{q}_{[t]}^\top \mathbf{w} + \lambda||\mathbf{w} - \hat{\mathbf{w}}_{[t]}||_1$, $f_c(\mathbf{w}) = -\mathbf{q}_{[t]}^\top \mathbf{w} + \lambda||\mathbf{w} - \hat{\mathbf{w}}_{[t]}||_1 + c(||\mathbf{w}||_1 - ||\mathbf{w}||_{(K)})$.

**Theorem 3.3.** *Let $L$ be the Lipschitz constant of $f$ on $\Delta_m$, then for all $c > 2\sqrt{2}L$, the following statements hold:*

*(i) $\mathbf{w}_c \in \Delta_m$ is a global optimal solution of problem (6) if and only if $\mathbf{w}_c$ is a global optimal solution of problem (8);*

*(ii) $\mathbf{w}_c \in \Omega$ is a local optimal solution of problem (6) if and only if $\mathbf{w}_c$ is a local optimal solution of problem (8).*

The proof of Theorem 3.3 is provided in appendix A.3. Theorem 3.3 tells us that problems (6) and (8) are equivalent in the sense that both problems share the same global (local) optimal solution sets provided that the penalty parameter $c$ is large enough. Therefore, we will design numerical algorithms solving problems (8) instead of solving problem (6).

### 3.2. Proximal Subgradient Algorithm

This section is devoted to the algorithm design for problem (8), which is a DC relaxation of $K$-sparse transaction cost optimization problem (6). We will first present the definition of a critical point of problem (8), and then establish the relationship between critical points and local minimizers of problem (8) and (6). Then, we characterize critical points of problem (8) by a fixed point equation. Lastly, a proximal subgradient algorithm is developed with the help of this characterization.

We begin with the definition of critical points of problem (8). The definition and calculus rules of the notion of subdifferential are presented in appendix A.1.

**Definition 3.4.** We say $\mathbf{w}^*$ is a critical point of problem (8) if $\mathbf{w}^* \in \Delta_m$ and the following inclusion holds for some $\mathbf{v} \in \partial||\cdot||_{(K)}(\mathbf{w}^*)$:

$$0 \in \partial f(\mathbf{w}^*) + \partial \iota_{\Delta_m}(\mathbf{w}^*) + c(\partial||\cdot||_1(\mathbf{w}^*) - \mathbf{v}),$$

where $\iota_{\Delta_m}(\mathbf{w})$ denotes the indicator function of $\Delta_m$, that is, $\iota_{\Delta_m}(\mathbf{w}) = 0$ if $\mathbf{w} \in \Delta_m$ and $\iota_{\Delta_m}(\mathbf{w}) = +\infty$ otherwise.

It is obvious from the generalized Fermat's Rule that a local minimizer of problem (8) must be a critical point of problem (8). Besides, we will illustrate in the following proposition that some critical points with some easily verified properties must be local minimizers of problem (8) or (6).

**Proposition 3.5.** *Let $\mathbf{w}^* \in \Delta_m$ be a critical point of problem (8). Then, the following statements hold:*

---

**Algorithm 1** PSGA for solving problem (8)

---

**Input**: parameters $c > 0$, $\alpha > 0$; tolerance $\epsilon$ and max iteration $k_{max}$.

1: Initialize: $\mathbf{w}^0 = (\frac{1}{m}, \cdots, \frac{1}{m})$, $k = 0$.

2: **while** $k \leq k_{max}$ or $\frac{||\mathbf{w}^{k+1} - \mathbf{w}^k||_2}{||\mathbf{w}^{k+1}||_2} > \epsilon$ **do**

3: $\quad \mathbf{v}^k \in \partial || \cdot ||_{(K)}(\mathbf{w}^k)$.

4:

$$\mathbf{w}^{k+1} = \text{prox}_{\alpha f + \iota_{\Delta_m}}(\mathbf{w}^k + \alpha c \mathbf{v}^k). \quad (11)$$

5: $\quad k = k + 1$.

6: **end while**

**Output**: Portfolio updates $\mathbf{w}_{[t+1]} = \mathbf{w}^{k+1}$.

---

(i) If $|\mathbf{w}^*_{(K)}| > |\mathbf{w}^*_{(K+1)}|$, then it is a local minimizer of problem (8).

(ii) If $||\mathbf{w}^*||_0 = K$, then $w^*$ is a local minimizer of problems (8) and (6).

The proof of Proposition 3.5 is provided in appendix A.4. We next characterize in the following theorem a critical point of problem (8) by a solution of a fixed point equation with the help of the notion of proximity operators. We first recall that for a proper closed convex function $g : \mathbb{R}^m \to (-\infty + \infty]$, its proximity operator, denoted by $\text{prox}_g$, is defined at $\mathbf{w} \in \mathbb{R}^m$ by

$$\text{prox}_g(\mathbf{w}) := \arg\min_{\mathbf{u} \in \mathbb{R}^m} \left( g(\mathbf{u}) + \frac{1}{2}||\mathbf{u} - \mathbf{w}||_2^2 \right). \quad (9)$$

**Theorem 3.6.** *The vector* $\mathbf{w}^*$ *is a critical point of problem (8) if and only if* $\mathbf{w}^*$ *satisfies the following fixed point equation for some* $\alpha > 0$ *and* $\mathbf{v} \in \partial || \cdot ||_{(K)}(\mathbf{w}^*)$:

$$\mathbf{w}^* = \text{prox}_{\alpha f + \iota_{\Delta_m}}(\mathbf{w}^* + \alpha c \mathbf{v}). \quad (10)$$

The proof of Theorem 3.6 is provided in appendix A.5. Inspired by Theorem 3.6, we develop a proximal subgradient algorithm (PSGA) for solving problem (8), which is stated in Algorithm 1.

### 3.3. ADMM for Solving Problem (11)

Note that the proximity operator of $\alpha f + \iota_{\Delta_m}$ has no closed form solution. We apply the alternating direction method of multipliers (ADMM) to compute $\text{prox}_{\alpha f + \iota_{\Delta_m}}$ in this section. The definition of proximity operator and (11) yield that:

$$(\mathbf{w}^{k+1}, \mathbf{w}^{k+1}) \in \arg\min_{\mathbf{w}, \mathbf{d} \in \mathbb{R}^m} g_1(\mathbf{w}) + g_2(\mathbf{d})$$
$$\text{s.t. } \mathbf{w} = \mathbf{d}, \quad (12)$$

where $g_1(\mathbf{w}) = -\mathbf{w}^\top(\mathbf{q}_{[t]} + c\mathbf{v}^k) + \frac{1}{2\alpha}||\mathbf{w} - \mathbf{w}^k||_2^2 + \iota_{\Delta_m}(\mathbf{w})$, $g_2(\mathbf{w}) = \lambda||\mathbf{w} - \hat{\mathbf{w}}_{[t]}||_1$. By this way, the augmented Lagrangian function of problem (12) is presented as:

$$L(\mathbf{w}, \mathbf{d}, \mu) = g_1(\mathbf{w}) + g_2(\mathbf{d}) + \mu^\top(\mathbf{w} - \mathbf{d}) + \frac{\rho}{2}||\mathbf{w} - \mathbf{d}||_2^2, \quad (13)$$

where $\mu$ is the Lagrangian multiplier, $\rho > 0$ controls the approximation of $\mathbf{d}$ to $\mathbf{w}$. Thus, ADMM generates the appropriate iteration for any arbitrary initial point $(\mathbf{w}^0, \mathbf{d}^0, \mu^0)$:

$$\begin{cases} \mathbf{w}^{k+1} \in \arg\min_{\mathbf{w} \in \mathbb{R}^m} L(\mathbf{w}, \mathbf{d}^k, \mu^k), & (14) \\ \mathbf{d}^{k+1} \in \arg\min_{\mathbf{d} \in \mathbb{R}^m} L(\mathbf{w}^{k+1}, \mathbf{d}, \mu^k), & (15) \\ \mu^{k+1} = \mu^k + \rho(\mathbf{w}^{k+1} - \mathbf{d}^{k+1}). & (16) \end{cases}$$

The following lemmas derive closed-form solutions of problems (14) and (15), and their proofs are provided in appendix A.6 and A.7, respectively.

**Lemma 3.7.** *The closed-form solution of the optimization problem (14) is*

$$\mathbf{w}^{k+1} = P_{\Delta_m}\left\{ \frac{1}{\frac{1}{\alpha} + \rho}(\frac{1}{\alpha}\mathbf{w}^k + \rho\mathbf{d}^k + \mathbf{q}_{[t]} + c\mathbf{v}^k - \mu^k) \right\},$$

*where* $P_{\Delta_m}$ *denotes the projection onto simplex* $\Delta_m$.

**Lemma 3.8.** *The closed-form solution of the optimization problem (15) is*

$$\mathbf{d}^{k+1} = \hat{\mathbf{w}}_{[t]} + \text{sign}(\phi)[|\phi| - \frac{\lambda}{\rho}]_+ ,$$

*where* $\phi = \mathbf{w}^{k+1} + \frac{1}{\rho}\mu^k - \hat{\mathbf{w}}_{[t]}$, *and* $([\mathbf{x}]_+)_i = max\{\mathbf{x}_i, 0\}$ *for* $i = 1, \cdots, m$, $\mathbf{x} \in \mathbb{R}^m_+$.

The detail procedure of PSGA-ADMM framework is summarized in Algorithm 2. We point out that $P_{\Delta_m}$ can be computed efficiently as shown in (Duchi et al., 2008).

We next establish in Theorem 3.9 the convergence of ADMM applied to solving problem (11) by showing that the corresponding Lagrangian function (13) has a saddle point. Its proof is provided in appendix A.8.

**Theorem 3.9.** *Let* $\{(\mathbf{w}^k, \mathbf{d}^k, \mu^k) : k \in \mathbb{N}\}$ *be generated by (14), (15) and (16). Then,* $\{\mathbf{w}^k - \mathbf{d}^k : k \in \mathbb{N}\}$ *converges to* $\mathbf{0}$, *and* $\{\mathbf{w}^k : k \in \mathbb{N}\}$ *converges to an optimal solution of problem (11).*

### 3.4. Convergence Analysis for PSGA

This subsection is devoted to the convergence analysis of PSGA. Although PSGA can be viewed as a proximal DC algorithm proposed in (Gotoh et al., 2018), the convergence

**Algorithm 2** PSGA-ADMM for the problem (8)

**Input**: parameters $c, \alpha, \lambda, \rho > 0$; window size $\omega$; tolerance $\epsilon$ and max iteration $k_{max}$.

1: Initialize: $\mathbf{w}^0 = (\frac{1}{m}, \cdots, \frac{1}{m})$, $k = 0$.
2: **while** $k \leq k_{max}$ **do**
3:  Initialize: $\mu^0 = (0, \cdots, 0)$, $j = 0$.
4:  Set $\mathbf{v} \in \partial \| \cdot \|_{(K)}(\mathbf{w}^k)$.
5:  **repeat**
6:    $\mathbf{w}^{j+1} = \mathrm{P}_{\Delta_m}\{ \frac{1}{\frac{1}{\alpha} + \rho}(\frac{1}{\alpha}\mathbf{w}^j + \rho \mathbf{d}^j + \mathbf{q}_{[t]} + c\mathbf{v} - \mu^j)\}$.
7:    $\mathbf{d}^{j+1} = \hat{\mathbf{w}}_{[t]} + \mathrm{sign}(\phi)[|\phi| - \frac{\lambda}{\rho}]_+$.
8:    $\mu^{j+1} = \mu^j + \rho(\mathbf{w}^{j+1} - \mathbf{d}^{j+1})$.
9:    $j = j + 1$.
10:  **until** $\frac{\|\mathbf{w}^{j+1} - \mathbf{w}^j\|_2}{\|\mathbf{w}^{j+1}\|_2} \leq \epsilon$ or $j > k_{max}$.
11:  $\mathbf{w}^{k+1} = \mathbf{w}^{j+1}$.
12:  **if** $\frac{\|\mathbf{w}^{k+1} - \mathbf{w}^k\|_2}{\|\mathbf{w}^{k+1}\|_2} \leq \epsilon$ **then**
13:    **return** $\mathbf{w}^{k+1}$ and break.
14:  **end if**
15:  $k = k + 1$.
16: **end while**

**Output**: Portfolio updates $\mathbf{w}_{[t+1]} = \mathbf{w}^{k+1}$.

of the entire sequence generated by it is not established in this literature. The global convergence of the sequence generated by extrapolation proximal DC algorithm (pDCA$_e$) for solving a DC problem is established in (Liu et al., 2019) by constructing an auxiliary function and showing it is a KL function. Motivated by pDCA$_e$, we construct a different auxiliary function which is more straightforward than that of pDCA$_e$. This auxiliary function will facilitate to establish the R-linear convergence rate of PSGA in this paper by showing its KL exponent is $1/2$. To this end, we first denote the objective function of problem (8) by $Q : \mathbb{R}^m \to (-\infty, +\infty]$, i.e.

$$Q(\mathbf{w}) := f(\mathbf{w}) + \iota_{\Delta_m} - c\|\mathbf{w}\|_{(K)}. \quad (17)$$

In order to show the convergence of the entire sequence generated by PSGA, we make use of the following auxiliary function and its KL property extensively in our analysis:

$$H(\mathbf{w}, \mathbf{v}) := f(\mathbf{w}) + \iota_{\Delta_m}(\mathbf{w}) + \iota_{S^*_{(K)}}(\mathbf{v}) - \langle \mathbf{w}, c\mathbf{v} \rangle, \quad (18)$$

where $S^*_{(K)}$ is the dual norm of the vector $K$-norm (Wu et al., 2014), i.e.,

$$S^*_{(K)} = \{\mathbf{z} \in \mathbb{R}^m : \|\mathbf{z}\|_{(K)^*} \leq 1\}$$
$$= \{\mathbf{z} \in \mathbb{R}^m : \|\mathbf{z}\|_\infty \leq 1, \|\mathbf{z}\|_1 \leq K\}.$$

It is clear that $\iota_{S^*_{(K)}}(\mathbf{v}) = (c\| \cdot \|_{(K)})^*(c\mathbf{v})$, where $(c\| \cdot \|_{(K)})^*$ is the conjugate function of $c\| \cdot \|_{(K)}$. Therefore, it yields from the Fenchel-Young inequality that for

any $\mathbf{w}, \mathbf{v} \in \mathbb{R}^m$,

$$H(\mathbf{w}, \mathbf{v}) \geq f(\mathbf{w}) + \iota_{\Delta_m}(\mathbf{w}) - c\|\mathbf{w}\|_{(K)} = Q(\mathbf{w}). \quad (19)$$

The next proposition illustrates that, for PSGA, the objective function values $Q$ as well as the auxiliary function values $H$ of the iterative sequence monotonically decreases, and the gap between the successive iterates vanishes.

**Proposition 3.10.** *Let $Q$ and $H$ be defined by (17) and (18), respectively. Let $\{(\mathbf{w}^k, \mathbf{v}^k) : k \in \mathbb{N}\}$ be generated by PSGA for any arbitrary $\mathbf{w}^0 \in \Delta_m$. Then, the following statements hold:*

*(i) For some $a > 0$, there has*

$$Q(\mathbf{w}^{k+1}) \leq H(\mathbf{w}^{k+1}, \mathbf{v}^k)$$
$$\leq Q(\mathbf{w}^k) - a\|\mathbf{w}^{k+1} - \mathbf{w}^k\|_2^2$$
$$\leq H(\mathbf{w}^k, \mathbf{v}^{k-1}) - a\|\mathbf{w}^{k+1} - \mathbf{w}^k\|_2^2;$$

*(ii) $\lim_{k\to\infty} \|\mathbf{w}^{k+1} - \mathbf{w}^k\|_2 = 0$;*

*(iii) $\lim_{k\to\infty} Q(\mathbf{w}^k)$ and $\lim_{k\to\infty} H(\mathbf{w}^{k+1}, \mathbf{v}^k)$ exist.*

The proof of Proposition 3.10 is provided in appendix A.9. In order to analyze the global convergence and convergence rate of PSGA, we require to establish the KL property and the KL exponent of $H$, respectively. The next proposition demonstrates that $H$ defined by (18) is a KL function and its KL exponent is $1/2$. The proof of Proposition 3.11 is placed in supplementary material A.10.

**Proposition 3.11.** *Let $H$ be defined by (18). Then $H$ is a KL function. Furthermore, the KL exponent of $H$ is $1/2$.*

With the help of Proposition 3.11, we establish in the following theorem the global convergence of the entire sequence generated by PSGA, through proving the sufficient descent condition and relative error condition in (Proposition 2.7, Li et al., 2022). The details of the proof are provided in supplementary material A.11.

**Theorem 3.12.** *Let $\{\mathbf{w}^k : k \in \mathbb{N}\}$ be generated by PSGA for any arbitrary $\mathbf{w}^0 \in \Delta_m$. Then we have:*

*(i) There exists $\mathbf{w}^* \in \Delta_m$ such that $\lim_{k\to\infty} \mathbf{w}^k = \mathbf{w}^*$ and $\mathbf{w}^*$ is a critical point of problem (8);*

*(ii) If $\mathbf{w}^*$ in Item (i) satisfies $|\mathbf{w}^*_{(K)}| > |\mathbf{w}^*_{(K+1)}|$, then $\{\mathbf{w}^k : k \in \mathbb{N}\}$ converges to a local minimizer of problem (8).*

*(iii) If $\mathbf{w}^*$ in Item (i) satisfies $\|\mathbf{w}^*\|_0 = K$, then $\{\mathbf{w}^k : k \in \mathbb{N}\}$ converges to a local minimizer of problem (6) and (8).*

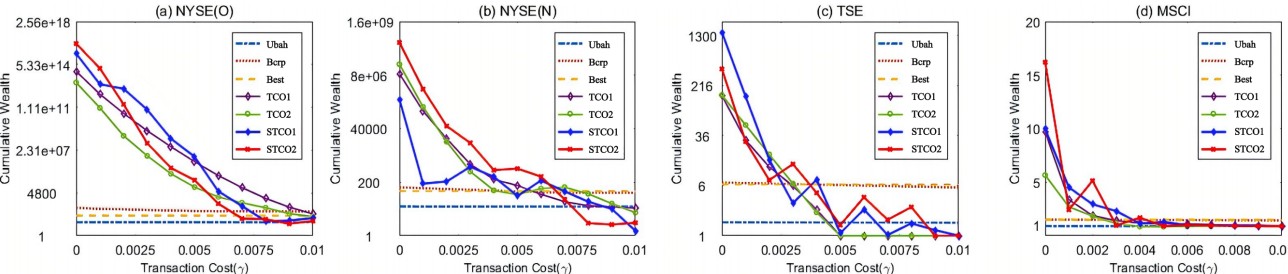

*Figure 1.* Cumulative wealth with varying transaction costs achieved by the proposed STCO1 and STCO2 in the four datasets. In order to emphasize the results under transaction costs and ensure clarity of the plot lines, we only present the comparison with benchmarks and TCO algorithms.

In addition to the sequential convergence of PSGA, we can also establish the $R$-linear convergence rate of PSGA since the KL exponent of $H$ is $1/2$. The main result is stated in the following theorem, and the detail of its proof is placed in appendix A.12.

**Theorem 3.13.** *Let* $\{\mathbf{w}^k : k \in \mathbb{N}\}$ *be generated by PSGA for any arbitrary* $\mathbf{w}^0 \in \Delta_m$. *Then,* $\{\mathbf{w}^k : k \in \mathbb{N}\}$ *converges $R$-linearly to a* $\mathbf{w}^*$. *Specifically, there exist* $a_0 > 0$, $b_0 \in (0, 1)$ *and* $k_0 > 0$ *such that*

$$\|\mathbf{w}^k - \mathbf{w}^*\|_2 \leq a_0 b_0^k \quad for \quad k \geq k_0.$$

## 4. Experimental Results

Extensive experiments with real-world financial datasets, which are NYSE(O) (Cover, 1991), NYSE(N) (Györfi et al., 2012; Li et al., 2013), TSE (Borodin et al., 2004) and MSCI (Li et al., 2012), are conducted to evaluate the performance of the proposed PSGA-ADMM for solving STCO. Moreover, we also consider three baseline methods: Ubah, Best and Bcrp (Cover, 1991), as well as 10 state-of-the-art methods: TCO1 (Li et al., 2017), TCO2 (Li et al., 2017), SSPO (Lai et al., 2018b), S1, S2 and S3 (Luo et al., 2020), DENRPO1-PARM, DENRPO1-OLMAR, DENRPO2-PARM and DENRPO2-OLMAR(Yao & Zhang, 2023a) as competitors in the experiments. Parameters of the proposed algorithm are set to: $c = 8e^{-6}$, $K = 10$, $\lambda = 10\gamma$, $\alpha = 0.0025$, $\rho = 0.618$, $\epsilon = 10^{-8}$, and $k_{max} = 10^8$. The detailed analysis of parameter sensitivity and the related experiments are presented in appendix B.1. Furthermore, we consider $\mathbf{q}_{[t]}$ in the following two cases according to the average reversion theory, $\mathbf{q}_{[t]1} := \frac{1}{\mathbf{x}_{[t]}}$ and $\mathbf{q}_{[t]2} := \frac{1}{w}\left(1 + \frac{1}{\mathbf{x}_{[t]}} + \cdots + \frac{1}{\odot_{i=0}^{w-2}\mathbf{x}_{[t-i]}}\right)$, called STCO1 and STCO2, respectively. For simplicity, the details of the datasets are placed in the appendix B.2, the competing algorithms and the source code of PSGA-ADMM are provided in appendix B.3.

### 4.1. Results for Cumulative Wealth with Transaction Cost

Cumulative wealth stands as one of the most critical metrics in investment analysis. We compare SSPO implementations with and without transaction cost algorithms to expose the importance of TCO and show the superiority of our research. As defined in (5), cumulative wealth with transaction cost is computed accordingly throughout the paper.

In order to highlight the main findings while keeping the presentation concise, we report summarized results in Table 1. The complete results are provided in Table 5 in Appendix B.5, which shows the cumulative wealth in fixed transaction cost rate $0\%$, $0.25\%$ and $0.5\%$. Firstly, we observe from Table 5 that comparing with benchmarks, the proposed methods outperform benchmarks in most cases, which lays the availability foundation. Moreover, strategies that ignore transaction costs (SSPO, S1, S2, and S3) may achieve higher returns in zero-cost scenarios. However, their performance degrades rapidly with increasing transaction costs, eventually dropping to zero even when transaction costs approach just $0.5\%$. In contrast, STCO maintains positive returns. As for strategies considering transaction costs, TCO preserves returns as transaction costs rise, while STCO consistently outperforms TCO in most cases. These results demonstrate the superiority of the proposed method in most cases under non-zero transaction costs.

Besides, Figure 1 reveals the cumulative wealth in varying transaction costs ranging from $0\%$ to $1\%$ comparing with TCO. Noting that the proposed STCO method always gains more wealth than TCO in most cases in a continuous transaction costs ascending process.

### 4.2. Results for Return and Risk Factors

In this subsection, we exhibit numerical results of three return and risk factors, which are Mean Excess Return (MER), $\alpha$ factor and $\beta$ factor. MER quantifies average excess return generated by a portfolio strategy relative to the Ubah benchmark. A higher MER indicates a superior portfolio strategy.

*Table 1.* Cumulative wealth with transaction costs from the proposed STCO1 and STCO2 and the compared algorithms in four data-sets, where UBAH is the benchmark, SSPO is the strategy without considering transaction costs, and TCOs are transaction cost optimization algorithms.

| | NYSE(O) | | | NYSE(N) | | |
|---|---|---|---|---|---|---|
| ALGORITHMS | 0% | 0.25% | 0.5% | 0% | 0.25% | 0.5% |
| UBAH | 14.50 | 14.46 | 14.43 | 18.06 | 18.01 | 17.97 |
| SSPO | **1.06E+18** | **2.45E+11** | 5.66E+04 | **1.62E+09** | 154.92 | 0.00 |
| TCO1 | 1.35E+14 | 5.57E+09 | **2.33E+06** | 9.15E+06 | **3.81E+03** | **143.47** |
| TCO2 | 1.47E+13 | 4.34E+07 | 1.52E+04 | 2.35E+07 | 2.14E+03 | 57.61 |
| **STCO1** | 6.16E+15 | **3.04E+11** | **6.54E+06** | 7.48E+05 | 274.18 | 53.88 |
| **STCO2** | **7.59E+16** | 1.24E+10 | 1.07E+05 | **3.46E+08** | **6.00E+05** | 271.82 |

| | TSE | | | MSCI | | |
|---|---|---|---|---|---|---|
| ALGORITHMS | 0% | 0.25% | 0.5% | 0% | 0.25% | 0.5% |
| UBAH | 1.61 | 1.61 | 1.60 | 0.91 | 0.90 | 0.90 |
| SSPO | **364.94** | **11.78** | 0.38 | 7.51 | 0.38 | 0.02 |
| TCO1 | 149 | 7.66 | 0.91 | 9.68 | 1.52 | **1.13** |
| TCO2 | 152.98 | **31.71** | **4.99** | 5.66 | 1.42 | 0.84 |
| **STCO1** | **1216.92** | 5.60 | 1.49 | **11.48** | **2.65** | **1.21** |
| **STCO2** | 95.92 | 3.97 | **4.88** | **15.17** | **2.05** | 0.76 |

Moreover, Sharpe (1964) shows that the intrinsic excess return, referred to as $\alpha$ factor in the financial industry, constitutes the part of expected return. An excellent algorithm has the higher $\alpha$ factor. Besides, $\beta$ factor (Sharpe, 1964) measures the risk of a portfolio strategy relative to the Ubah benchmark. A value of $\beta > 0$ ($\beta < 0$) signifies a positive (negative) correlation between the strategy return and the Ubah benchmark return. Furthermore, $|\beta| < 1$ ($|\beta| > 1$) implies that the strategy return exhibits lower (higher) volatility compared to the Ubah benchmark. The detail for computing MER, $\alpha$ and $\beta$ factor are shown in appendix B.4.

Table 6 placed in appendix B.5 show above return and risk factors of STCO comparing with TCO and DENRPO strategies in transaction cost rate $\gamma = 0\%$, $\gamma = 0.25\%$ and $\gamma = 0.5\%$. It is obviously that STCO achieves greater MER and $\alpha$ factor as well as smaller $\beta$ factor in most cases. This verifies that the sparsity of the portfolio vector considered in the proposed model decreases the risk of the model while maintaining a high return.

### 4.3. Sparsity for STCO

In this subsection, we examine the sparsity of the portfolios generated by STCO. Sparsity can be measured by the number of nonzero elements of the portfolio vector $w$. The results are reported in Table 7 in appendix B.6, which presents the average sparsity and the corresponding standard deviation of the portfolios generated by STCO1 and STCO2 and other state-of-the-art algorithms with sparsity level $K = 10$ in transaction rate $\gamma = 0\%$ and $\gamma = 0.25\%$. We observe that the average sparsity of STCO1 is about 1.0531 on NYSE(O) when $\gamma = 0\%$, and similar values can also be observed across other datasets and settings. This indicates that the

proposed STCO method selects only a small number of assets while maintaining strong portfolio performance.

## 5. Concluding Remarks

In this paper, we propose a $K$-sparse TCO model which considers the transaction cost in the online portfolio selection and the sparsity of the asset vector simultaneously, and develop an R-linearly convergent algorithm to solve it. Numerical experiments demonstrate that benefiting from the sparsity term, the proposed model decreases the risk of the model while maintaining a high return. Noting the significant impact of the sparsity, future work will incorporate the deep learning technique to learn a more adaptive data-driven sparsity promoting neural network to enhance the performance of the sparse TCO model in various investment markets.

## Acknowledgements

This research was funded by the National Science Foundation of China grant number 12271181 and by the Guangzhou Basic Research Program grant number 2025A04J5240.

## Impact Statement

This paper presents work whose goal is to advance the field of Machine Learning. There are many potential societal consequences of our work, none which we feel must be specifically highlighted here.

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

# A. Theoretical Analysis

## A.1. Notation and Preliminaries

We begin with our preferred notations. We denote by $\mathbb{R}^n$ the Euclidean space of dimension $n$, and denote $\mathcal{B}(x, \delta)$ as an open ball centered at $x \in \mathbb{R}^n$ with radius $\delta > 0$, i.e., $\mathcal{B}(x, \delta) := \{z \in \mathbb{R}^n : \|x - z\|_2 < \delta\}$. The Cartesian product of the sets $\mathcal{A}_1$ and $\mathcal{A}_2$ is denoted by $\mathcal{A}_1 \times \mathcal{A}_2$. The distance from a vector $x \in \mathbb{R}^n$ to a set $\mathcal{A} \subseteq \mathbb{R}^n$ is denoted by $\mathrm{dist}(x, \mathcal{A}) := \inf\{\|x - y\|_2 : y \in \mathcal{A}\}$. The indicator function on a nonempty set $\mathcal{A} \subseteq \mathbb{R}^n$ is defined by

$$\iota_{\mathcal{A}}(x) = \begin{cases} 0, & \text{if } x \in \mathcal{A}, \\ +\infty, & \text{else.} \end{cases}$$

An extended-real-valued function $\varphi : \mathbb{R}^n \to (-\infty, +\infty]$ is said to be proper if its domain $\mathrm{dom}\,\varphi := \{x \in \mathbb{R}^n : \varphi(x) < +\infty\}$ is nonempty. A proper function $\varphi$ is said to be closed if $\varphi$ is lower semi-continuous on $\mathbb{R}^n$. A function $\varphi : \mathbb{R}^n \to (-\infty, +\infty]$ is said to be Lipschitz continuous on a set $\mathcal{A}$ with a modulus $L > 0$, if there exists $\delta > 0$ such that

$$\|\varphi(u) - \varphi(v)\|_2 \le L \|u - v\|_2$$

holds for all $u, v \in \mathcal{B}(x, \delta) \cap \mathcal{A}$.

We next review some preliminaries on subdifferentials of nonconvex functions (Cui & Pang, 2021; Mordukhovich, 2006; Rockafellar & Wets, 1998). For a proper function $\varphi$, the Fréchet subdifferential at $x \in \mathrm{dom}\,\varphi$ is defined by

$$\widehat{\partial}\varphi(x) := \left\{ y \in \mathbb{R}^n : \liminf_{\substack{z \to x \\ z \neq x}} \frac{\varphi(z) - \varphi(x) - \langle y, z - x \rangle}{\|z - x\|_2} \ge 0 \right\},$$

and the limiting subdifferential at $x \in \mathrm{dom}\,\varphi$ is defined by

$$\partial\varphi(x) := \left\{ y \in \mathbb{R}^n : \exists x^k \to x, \varphi(x^k) \to \varphi(x), y^k \to y \text{ with } y^k \in \widehat{\partial}\varphi(x^k) \text{ for each } k \right\}.$$

We set $\widehat{\partial}\varphi(x) = \partial\varphi(x) = \varnothing$ by convention when $x \notin \mathrm{dom}\,\varphi$, and define $\mathrm{dom}\,\partial\varphi := \{x : \partial\varphi(x) \neq \varnothing\}$. For a convex $\varphi$, the Fréchet and limiting subdifferential both consist with the classical subdifferential of a convex function at each $x \in \mathrm{dom}\,\varphi$ (Rockafellar and Wets, 1998, Proposition 8.12), that is,

$$\widehat{\partial}\varphi(x) = \partial\varphi(x) = \{y \in \mathbb{R}^n : \varphi(z) - \varphi(x) - \langle y, z - x \rangle \ge 0, \text{ for all } z \in \mathbb{R}^n\}.$$

It is clear that $\partial(\alpha\varphi)(x) = \alpha\partial\varphi(x)$ for all $\alpha > 0$ and $x \in \mathrm{dom}\,\varphi$. For proper lower semicontinuous functions $\varphi_1, \varphi_2 : \mathbb{R}^n \to (-\infty, +\infty]$, there holds $\partial(\varphi_1 + \varphi_2)(x) \subseteq \partial\varphi_1(x) + \partial\varphi_2(x)$ if $\varphi_1$ is locally Lipschitz continuous around $x$. If $\varphi_1$ and $\varphi_2$ are further convex, we have $\partial(\varphi_1 - \varphi_2)(x) \subseteq \partial\varphi_1(x) - \partial\varphi_2(x)$ for $x \in \mathrm{dom}\,\partial\varphi_2$. For a function $\varphi : \mathbb{R}^n \to (-\infty, +\infty]$, the generalized *Fermat's Rule* states that $0 \in \partial\varphi(x^*)$ holds if $x^* \in \mathrm{dom}(\varphi)$ is a local minimizer of $\varphi$ (Rockafellar and Wets, 1998, Theorem 10.1).

The convex conjugate function $\varphi^* : \mathbb{R}^n \to [-\varphi(0), +\infty]$ is defined at $y \in \mathbb{R}^n$ as $\varphi^*(y) := \sup\{\langle x, y \rangle - \varphi(x) : x \in \mathbb{R}^n\}$. The conjugate $\varphi^*$ is a proper closed convex function (Rockafellar, 1970, Theorem 12.2). Furthermore, for any $x, y \in \mathbb{R}^n$, the following three statements are equivalent (Rockafellar and Wets, 1998, Proposition 11.3):

$$\langle x, y \rangle = \varphi(x) + \varphi^*(y) \Leftrightarrow y \in \partial\varphi(x) \Leftrightarrow x \in \partial\varphi^*(y).$$

Therefore, it is clear that the *Fenchel-Young Inequality*, i.e., $\varphi(x) + \varphi^*(y) \ge \langle x, y \rangle$, holds for all $x, y \in \mathbb{R}^n$, and it becomes an equality when $y \in \partial\varphi(x)$. Besides, from the *Fenchel-Young Inequality*, we know that $y \in \mathrm{dom}\,\varphi^*$ if $y \in \partial\varphi(x)$ for some $x \in \mathbb{R}^n$.

## A.2. Proof of Proposition 3.2

*Proof.* For $\forall \epsilon > 0$, take $\mathbf{z} \in \Omega, \bar{\mathbf{w}} \in \mathcal{B}(\mathbf{z}, \epsilon) \cap \Delta_m$. If $\bar{\mathbf{w}} \in \Omega$, then we can get $\mathrm{dist}(\bar{\mathbf{w}}, \Omega) = 0$ and $p(\bar{\mathbf{w}}) = 0$ to end the proof. Thus, we can finish the proof if and only if for any $\bar{\mathbf{w}} \in [\mathcal{B}(\mathbf{z}, \epsilon) \cap \Delta_m] \setminus \Omega$ satisfies the condition.

In this case, we can take $\bar{\mathbf{w}} \in [\mathcal{B}(\mathbf{z}, \epsilon) \cap \Delta_m] \setminus \Omega$, thus $p(\bar{\mathbf{w}}) > 0$ and $\mathrm{dist}(\bar{\mathbf{w}}, \Omega) > 0$. For any $\mathbf{w} \in \Delta_m$, let

$$h(\mathbf{w}) = p(\mathbf{w}) + \frac{2p(\bar{\mathbf{w}})}{\mathrm{dist}(\bar{\mathbf{w}}, \Omega)}||\mathbf{w} - \bar{\mathbf{w}}||,$$

it can be proofed by the definition that $h(\mathbf{w})$ is a lower semi-continuous function in the closed set. Thus, we can assume $\mathbf{y} \in \Delta_m$ is the optimal solution of $\arg\min_{\mathbf{w} \in \Delta_m} h(\mathbf{w})$, then we have

$$h(\mathbf{y}) = p(\mathbf{y}) + \frac{2p(\bar{\mathbf{w}})}{\mathrm{dist}(\bar{\mathbf{w}}, \Omega)}||\mathbf{y} - \bar{\mathbf{w}}|| \leq h(\mathbf{w}), \tag{20}$$

further derived from (20) that

$$0 \leq p(\mathbf{y}) \leq p(\mathbf{w}) + \frac{2p(\bar{\mathbf{w}})}{\mathrm{dist}(\bar{\mathbf{w}}, \Omega)}||\mathbf{w} - \mathbf{y}||. \tag{21}$$

Substitute $\mathbf{w} = \bar{\mathbf{w}}$ into (21) and the inequality can be deduced to

$$||\mathbf{y} - \bar{\mathbf{w}}|| \leq \frac{1}{2}\mathrm{dist}(\bar{\mathbf{w}}, \Omega) < \mathrm{dist}(\bar{\mathbf{w}}, \Omega) \leq \epsilon. \tag{22}$$

Therefore, $\mathbf{y} \notin \Omega$ and we can get $p(\mathbf{y}) > 0$ and $||\mathbf{y}||_0 > k$, that is $\mathbf{y}_{(K)}, \mathbf{y}_{(K+1)} > 0$. Let $\bar{i}$ and $\bar{j}$ denote the index of $K$-th and $(K+1)$-th largest element of $\mathbf{y}$, respectively. Take $\hat{\mathbf{y}} = \mathbf{y} - \mathbf{y}_j(e_j - e_i) \in \Delta_m$, then we have

$$||\hat{\mathbf{y}} - \mathbf{y}|| = ||\mathbf{y}_j(e_j - e_i)|| = \sqrt{2}\mathbf{y}_j, \tag{23}$$

thus,

$$p(\mathbf{y}) - p(\hat{\mathbf{y}}) = ||\hat{\mathbf{y}}||_{(K)} - ||\mathbf{y}||_{(K)} = \mathbf{y}_j > 0. \tag{24}$$

Substitute $\mathbf{w} = \hat{\mathbf{y}}$ into (21) and then it leads to:

$$p(\mathbf{y}) - p(\hat{\mathbf{y}}) \leq \frac{2p(\bar{\mathbf{w}})}{\mathrm{dist}(\bar{\mathbf{w}}, \Omega)}||\hat{\mathbf{y}} - \mathbf{y}|| = \frac{2\sqrt{2}p(\bar{\mathbf{w}})}{\mathrm{dist}(\bar{\mathbf{w}}, \Omega)}\mathbf{y}_j. \tag{25}$$

Thus $\mathrm{dist}(\bar{\mathbf{w}}, \Omega) \leq 2\sqrt{2}p(\bar{\mathbf{w}})$, then we finish the proof. $\qquad\square$

### A.3. Proof of Theorem 3.3

*Proof.* We first prove Item (i). For the sufficiency. If $p(\mathbf{w}_c) > 0$, then for all $\mathbf{w}, \mathbf{w}_c \in \Delta_m$,

$$\begin{aligned}
f_c(\mathbf{w}_c) &= f(\mathbf{w}_c) + cp(\mathbf{w}_c) \\
&> f(\mathbf{w}_c) + 2\sqrt{2}Lp(\mathbf{w}_c) \\
&\geq f(\mathbf{w}_c) + L||\mathbf{w}_c - \Pi_\Omega(\mathbf{w})|| \\
&\geq f(\Pi_\Omega(\mathbf{w})) \\
&= f_c(\Pi_\Omega(\mathbf{w})),
\end{aligned} \tag{26}$$

where the second inequality is found by Proposition 3.2, the last inequality is held since $f(\mathbf{w})$ is Lipschitz continuous on $\Delta_m$, and the last equality is deduced from $p(\Pi_\Omega(\mathbf{w})) = 0$. Thus, $f_c(\mathbf{w}_c) > f_c(\Pi_\Omega(\mathbf{w}))$, which conflicts with that $\mathbf{w}_c$ is the global optimal solution of problem (8). Therefore, $p(\mathbf{w}_c) = 0$, that is $\mathbf{w}_c \in \Omega$ and $f_c(\mathbf{w}) = f(\mathbf{w})$. Then for all $\mathbf{w}, \mathbf{w}_c \in \Delta_m$,

$$f(\mathbf{w}_c) = f_c(\mathbf{w}_c) \leq f_c(\mathbf{w}) = f(\mathbf{w}), \tag{27}$$

implying that $\mathbf{w}_c \in \Delta_m$ is the global optimal solution of problem (6).

For the necessity. If $\mathbf{w}_c \in \Delta_m$ is the global optimal solution of problem (6) then for all $\mathbf{w} \in \Delta_m$, we have

$$\begin{aligned}
f_c(\mathbf{w}_c) = f(\mathbf{w}_c) &\leq f(\Pi_\Omega(\mathbf{w})) \\
&\leq f(\mathbf{w}) + L||\mathbf{w} - \Pi_\Omega(\mathbf{w})|| \\
&\leq f(\mathbf{w}) + 2\sqrt{2}Lp(\mathbf{w}) \\
&< f(\mathbf{w}) + cp(\mathbf{w}) \\
&= f_c(\mathbf{w}),
\end{aligned} \tag{28}$$

where the second inequality is held since $f(\mathbf{w})$ is Lipschitz continuous. Thus, $f_c(\mathbf{w}_c) \leq f_c(\mathbf{w})$, indicates that $\mathbf{w}_c$ is the global optimal solution of problem (8). Combining the sufficiency and the necessity, we finish the proof.

We next show Item (ii). For the necessity. If $\mathbf{w}_c \in \Omega$ is a local optimal solution of problem (6) then for all $\mathbf{w} \in \mathcal{B}(\mathbf{w}_c, \epsilon)$ and $\epsilon > 0$, there has $p(\mathbf{w}_c) = 0$ and $f(\mathbf{w}_c) \leq f(\mathbf{w})$. Since $p(\mathbf{w})$ is continuous, there exists $\hat{\epsilon} > 0$ such that $||p(\mathbf{w}) - p(\mathbf{w}_c)|| = p(\mathbf{w}) \leq \frac{\epsilon}{4\sqrt{2}}$ for all $\mathbf{w} \in \mathcal{B}(\mathbf{w}_c, \hat{\epsilon})$. Take $0 < \tilde{\epsilon} \leq \min(\frac{\epsilon}{2}, \hat{\epsilon})$, then for all $\mathbf{w} \in \mathcal{B}(\mathbf{w}_c, \tilde{\epsilon})$,

$$\text{dist}(\mathbf{w}, \Omega) = ||\mathbf{w} - \Pi_\Omega(\mathbf{w})|| \leq 2\sqrt{2}p(\mathbf{w}) \leq \frac{\epsilon}{2}. \tag{29}$$

Thus, it indicates that

$$||\mathbf{w}_c - \Pi_\Omega(\mathbf{w})|| \leq ||\mathbf{w} - \Pi_\Omega(\mathbf{w})|| + ||\mathbf{w} - \mathbf{w}_c|| \leq \frac{\epsilon}{2} + \tilde{\epsilon} < \epsilon, \tag{30}$$

that is $\Pi_\Omega(\mathbf{w}) \in \mathcal{B}(\mathbf{w}_c, \epsilon) \cap \Omega$. Considering Proposition 3.2 and (29), we have

$$\begin{aligned} f_c(\mathbf{w}_c) = f(\mathbf{w}_c) &\leq f(\Pi_\Omega(\mathbf{w})) \\ &\leq f(\mathbf{w}) + L||\mathbf{w} - \Pi_\Omega(\mathbf{w})|| \\ &\leq f(\mathbf{w}) + 2\sqrt{2}Lp(\mathbf{w}) \\ &< f(\mathbf{w}) + cp(\mathbf{w}) \\ &= f_c(\mathbf{w}), \end{aligned} \tag{31}$$

thus $\mathbf{w}_c$ is a local optimal solution of problem (8).

For the sufficiency. If $\mathbf{w}_c \in \Omega$ is a local optimal solution of problem (8), then $p(\mathbf{w}_c) = 0$. Therefore, for all $\mathbf{w} \in \mathcal{B}(\mathbf{w}_c, \epsilon) \cap \Omega$,

$$f(\mathbf{w}_c) = f_c(\mathbf{w}_c) \leq f_c(\mathbf{w}) = f(\mathbf{w}), \tag{32}$$

that is $\mathbf{w}_c$ is a local optimal solution of problem (6). $\square$

## A.4. Proof of Proposition 3.5

*Proof.* We first prove Item (i). Let $W^* := \{\mathbf{w}^*_{(1)}, \cdots, \mathbf{w}^*_{(K)}\}$. Since $|\mathbf{w}^*_{(K)}| > |\mathbf{w}^*_{(K+1)}|$, we have

$$\partial \|\cdot\|_{(K)}(\mathbf{w}^*) = \{(\mathbf{v}_1, \mathbf{v}_2, \cdots, \mathbf{v}_m) : \mathbf{v}_i = \text{sign}(\mathbf{w}^*_i) \text{ if } \mathbf{w}^*_i \in W^* \text{ and } \mathbf{v}_i = 0 \text{ otherwise}\}.$$

Thus, it yields that

$$\begin{aligned} \partial \|\cdot\|_1(\mathbf{w}^*) - \partial \|\cdot\|_{(K)}(\mathbf{w}^*) = \{&(\mathbf{v}_1, \mathbf{v}_2, \cdots, \mathbf{v}_m) : \mathbf{v}_i = 0 \text{ if } \mathbf{w}^*_i \in W^*, \mathbf{v}_i = \text{sign}(\mathbf{w}^*_i) \text{ if } 0 \neq \mathbf{w}^*_i \notin W^* \\ &\text{and } \mathbf{v}_i \in [-1, 1] \text{ if } \mathbf{w}^*_i = 0\}. \end{aligned} \tag{33}$$

Without loss of generality, we suppose $\mathbf{w}^*_i = \mathbf{w}^*_{(i)}$ for $i \in \{1, 2, \cdots, K\}$. Then, $\mathbf{w}^*$ is a critical point of problem (8) is equivalent to

$$0 \in \partial f(\mathbf{w}^*) + \partial \iota_{\Delta_m}(\mathbf{w}^*) + c\partial \sum_{i=K+1}^{m} |\mathbf{w}_i|(\mathbf{w}^*), \tag{34}$$

since $\partial \sum_{i=K+1}^{m} |\mathbf{w}_i|(\mathbf{w}^*) = \partial \|\cdot\|_1(\mathbf{w}^*) - \partial \|\cdot\|_{(K)}(\mathbf{w}^*)$ due to (33). Noting that $f(\mathbf{w}) + \iota_{\Delta_m}(\mathbf{w}) + \sum_{i=K+1}^{m} |\mathbf{w}_i|$ is a convex function, we deduce from (34) that

$$\mathbf{w}^* \in \arg\min \; f(\mathbf{w}) + c \sum_{i=K+1}^{m} |\mathbf{w}_i| : \mathbf{w} \in \Delta_m. \tag{35}$$

Invoking $|\mathbf{w}^*_{(K)}| > |\mathbf{w}^*_{(K+1)}|$, there exists $\delta > 0$ such that for any $\mathbf{w} \in \mathcal{B}(\mathbf{w}^*, \delta)$ there holds

$$\|\mathbf{w}\|_{(K)} = \sum_{i=1}^{K} |\mathbf{w}_i|. \tag{36}$$

Therefore, (35) leads to that for any $\mathbf{w} \in \mathcal{B}(\mathbf{w}^*, \delta) \cap \Delta_m$,

$$f(\mathbf{w}^*) + \iota_{\Delta_m}(\mathbf{w}^*) + c \sum_{i=K+1}^{m} |\mathbf{w}_i^*| \leq f(\mathbf{w}) + \iota_{\Delta_m}(\mathbf{w}) + c \sum_{i=K+1}^{m} |\mathbf{w}_i| = f(\mathbf{w}) + \iota_{\Delta_m}(\mathbf{w}) + c(\|\mathbf{w}\|_1 - \|\mathbf{w}\|_{(K)}), \tag{37}$$

where the last equality comes from (36). In view of $\|\mathbf{w}^*\|_{(K)} = \sum_{i=1}^{K} |\mathbf{w}_i^*|$. (37) indicates that $\mathbf{w}^*$ is a local minimizer of problem (8).

We next show Item (ii). Item (i) implies that $\mathbf{w}^*$ is a local minimizer of problem (8). We next dedicate to proving it is also a local minimizer of problem (6). Without loss of generality, we suppose $\mathrm{supp}(\mathbf{w}^*) = \{1, 2, \cdots, K\}$. Then, similar with the previous proof, (34) and (35) hold.

Due to $\|\mathbf{w}^*\|_0 = K$, there exists $\delta > 0$ such that

$$\mathrm{supp}(\mathbf{w}) = \{1, 2, \ldots, K\} \quad \text{for all} \ \ \mathbf{w} \in \mathcal{B}(\mathbf{w}^*, \delta) \cap \{\mathbf{w} \in \Delta_m : \|\mathbf{w}\|_0 \leq K\}.$$

Then, (35) together with $\mathrm{supp}(\mathbf{w}^*) = \mathrm{supp}(\mathbf{w}) = \{1, 2, \ldots, K\}$ yields that for all $\mathbf{w} \in \mathcal{B}(\mathbf{w}^*, \delta) \cap \{\mathbf{w} \in \Delta_m : \|\mathbf{w}\|_0 \leq K\}$, there holds

$$f(\mathbf{w}^*) = f(\mathbf{w}^*) + c \sum_{i=K+1}^{m} |\mathbf{w}_i^*| \leq f(\mathbf{w}) + c \sum_{i=K+1}^{m} |\mathbf{w}_i| = f(\mathbf{w}).$$

This indicates that $\mathbf{w}^*$ is a local minimizer of problem (6).

$\square$

## A.5. Proof of Theorem 3.6

*Proof.* From the definition of proximity operators (9), (10) is equivalent to

$$\mathbf{w}^* = \arg\min_{\mathbf{w} \in \Delta_m} \ \alpha f(\mathbf{w}) + \frac{1}{2}\|\mathbf{w} - \mathbf{w}^* - \alpha c\mathbf{v}\|_2^2,$$

which can be rewritten as

$$\mathbf{w}^* = \arg\min_{\mathbf{w} \in \Delta_m} \ \alpha f(\mathbf{w}) + c\|\mathbf{w}\|_1 + \frac{1}{2}\|\mathbf{w} - \mathbf{w}^* - \alpha c\mathbf{v}\|_2^2, \tag{38}$$

due to $\|\mathbf{w}\|_1 = 1$ for all $\mathbf{w} \in \Delta_m$.

Noting that (38) is a convex optimization problem, by applying the generalized Fermat's Rule to (38), we obtain that (38) is equivalent to that $\mathbf{w}^*$ is a critical point of problem 8. We then complete the proof immediately.

$\square$

## A.6. Proof of Lemma 3.7

*Proof.* We simply (14) as follows:

$$\begin{aligned} \mathbf{w}^{k+1} &\in \arg\min_{\mathbf{w} \in \Delta_m} \frac{1}{2\alpha}\|\mathbf{w} - \mathbf{w}^k\|_2^2 + \frac{\rho}{2}\|\mathbf{w} - \mathbf{d}^k\|_2^2 + <\mathbf{w}, \mu^k - \mathbf{q}_{[t]} - c\mathbf{v}^k> \\ &= \arg\min_{\mathbf{w} \in \Delta_m} \frac{1}{2}(\frac{1}{\alpha} + \rho)\|\mathbf{w} - \frac{1}{\frac{1}{\alpha} + \rho}(\frac{1}{\alpha}\mathbf{w}^k + \rho\mathbf{d}^k + \mathbf{q}_{[t]} + c\mathbf{v}^k - \mu^k\|_2^2) \\ &= \mathrm{P}_{\Delta_m}\{\frac{1}{\frac{1}{\alpha} + \rho}(\frac{1}{\alpha}\mathbf{w}^k + \rho\mathbf{d}^k + \mathbf{q}_{[t]} + c\mathbf{v}^k - \mu^k)\}. \end{aligned}$$

$\square$

### A.7. Proof of Lemma 3.8

*Proof.* Problem (15) can be reformulated as:

$$
\begin{aligned}
\mathbf{d}^{k+1} \in \arg\min_{\mathbf{d}\in\mathbb{R}^m} & \ \lambda||\mathbf{d} - \hat{\mathbf{w}}_{[t]}||_1 + \frac{\rho}{2}||\mathbf{w}^{k+1} - \mathbf{d}||_2^2 - <\mathbf{d}, \mu^k> \\
= \arg\min_{\mathbf{d}\in\mathbb{R}^m} & \ \lambda||\mathbf{d} - \hat{\mathbf{w}}_{[t]}||_1 + \frac{\rho}{2}||\mathbf{d} - \frac{1}{\rho}(\rho\mathbf{w}^{k+1} + \mu^k)||_2^2.
\end{aligned}
\tag{39}
$$

Let $\mathbf{z} = \mathbf{d} - \hat{\mathbf{w}}_{[t]}$, then (39) is equivaluated to

$$
\begin{aligned}
\mathbf{z}^{k+1} \in \arg\min_{\mathbf{z}\in\mathbb{R}^m} & \ \frac{\lambda}{\rho}||\mathbf{z}||_1 + \frac{1}{2}||\mathbf{z} - \frac{1}{\rho}(\rho\mathbf{w}^{k+1} + \mu^k - \rho\hat{\mathbf{w}}_{[t]})||_2^2 \\
= \arg\min_{\mathbf{z}\in\mathbb{R}^m} & \ \frac{\lambda}{\rho}||\mathbf{z}||_1 + \frac{1}{2}||\mathbf{z} - \phi||_2^2 \\
= & \ \text{sign}(\phi)[|\phi| - \frac{\lambda}{\rho}]_+,
\end{aligned}
$$

where $\phi = \frac{1}{\rho}(\rho\mathbf{w}^{k+1} + \mu^k - \rho\hat{\mathbf{w}}_{[t]})$. Hence, the closed-form solution of $\mathbf{d}$ is

$$
\mathbf{d}^{k+1} = \hat{\mathbf{w}}_{[t]} + \mathbf{z}^{k+1} = \hat{\mathbf{w}}_{[t]} + \text{sign}(\phi)[|\phi| - \frac{\lambda}{\rho}]_+.
$$

$\square$

### A.8. Proof of Theorem 3.9

*Proof.* We can find in (Bertsekas, 2015) that the convergence of the ADMM is guaranteed when the Lagrangian function associated with the optimization problem admits a saddle point. Thus, we delicate to demonstrate in the following that Lagrangian function (13) has a saddle point to establish the convergence of ADMM applied for problem (11).

We first show that the optimal solution set of problem (11) is nonempty. It is obvious that $f + \iota_{\Delta_m}$ is convex, then its corresponding proximity operator problem $\text{prox}_{f+\iota_{\Delta_m}}$ is a strong convex problem. This indicates problem (11) exist an optimal solution. We suppose $\mathbf{w}^*$ is an optimal solution of problem (11). That is $\mathbf{w}^* \in \arg\min g_1(\mathbf{w}) + g_2(\mathbf{w})$. Then, the Fermat's rule leads to $\mathbf{0} \in \partial(g_1 + g_2)(\mathbf{w}^*)$. Since $g_1, g_2$ are convex and

$$
\text{dom}(g_2) := \{\mathbf{x} \in \mathbb{R}^m : g_2(\mathbf{x}) < +\infty\} = \mathbb{R}^m,
$$

we have $\mathbf{0} \in \partial g_1(\mathbf{w}^*) + \partial g_2(\mathbf{w}^*)$. Thus, there exists

$$
\mu^* \in \partial g_2(\mathbf{w}^*) \text{ such that } -\mu^* \in \partial g_1(\mathbf{w}^*).
$$

Set $\mathbf{d}^* = \mathbf{w}^*$. We next show $(\mathbf{w}^*, \mathbf{d}^*, \mu^*)$ is a saddle point of $L(\mathbf{w}, \mathbf{d}, \mu)$.

First, it is obvious that

$$
\begin{cases}
\mathbf{0} \in \partial g_1(\mathbf{w}^*) + \mu^* + \rho(\mathbf{w}^* - \mathbf{d}^*), \\
\mathbf{0} \in \partial g_2(\mathbf{d}^*) - \mu^* + \rho(\mathbf{d}^* - \mathbf{w}^*),
\end{cases}
$$

which leads to $(\mathbf{w}^*, \mathbf{d}^*)$ is a minimizer of $L(\mathbf{w}, \mathbf{d}, \mu^*)$ due to the convexity of $L(\mathbf{w}, \mathbf{d}, \mu^*)$. This implies that

$$
L(\mathbf{w}^*, \mathbf{d}^*, \mu^*) \le L(\mathbf{w}, \mathbf{d}, \mu^*)
$$

for all $\mathbf{w}, \mathbf{d} \in \mathbb{R}^m$. Second, it is easy to see

$$
L(\mathbf{w}^*, \mathbf{d}^*, \mu^*) \ge L(\mathbf{w}^*, \mathbf{d}^*, \mu)
$$

for any $\mu \in \mathbb{R}^m$, since $\mathbf{w}^* = \mathbf{d}^*$.

Therefore, we can obtain that the Lagrangian function (13) has a saddle point, i.e., there exists $(\mathbf{w}^*, \mathbf{d}^*, \mu^*)$ such that

$$
L(\mathbf{w}^*, \mathbf{d}^*, \mu) \le L(\mathbf{w}^*, \mathbf{d}^*, \mu^*) \le L(\mathbf{w}, \mathbf{d}, \mu^*)
$$

for all $\mathbf{w}, \mathbf{d}, \mu \in \mathbb{R}^m$. This together with Proposition 5.4.1 in (Bertsekas, 2015) leads to the convergence of ADMM. We complete the proof. $\square$

## A.9. Proof of Proposition 3.10

*Proof.* Since $Q$ is lower bounded, Items (ii) and (iii) of this proposition are direct consequences of Item (i). Thus, we dedicate to showing Item (i) in the remaining part of this proof.

We deduce from the definition of proximity operators and (10) that

$$\alpha f(\mathbf{w}^{k+1}) + \iota_{\Delta_m}(\mathbf{w}^{k+1}) + \frac{1}{2}\|\mathbf{w}^{k+1} - \mathbf{w}^k - \alpha c\mathbf{v}^k\|_2^2 \leq \alpha f(\mathbf{w}^k) + \iota_{\Delta_m}(\mathbf{w}^k) + \frac{1}{2}\|\alpha c\mathbf{v}^k\|_2^2,$$

which implies that

$$f(\mathbf{w}^{k+1}) + \iota_{\Delta_m}(\mathbf{w}^{k+1}) + \frac{1}{2\alpha}\|\mathbf{w}^{k+1} - \mathbf{w}^k\|_2^2 - \langle \mathbf{w}^{k+1} - \mathbf{w}^k, c\mathbf{v}^k \rangle \leq f(\mathbf{w}^k) + \iota_{\Delta_m}(\mathbf{w}^k). \tag{40}$$

On the other hand, invoking $\iota_{S_{(K)}^*}(\mathbf{v}) = (c\|\cdot\|_{(K)})^*(c\mathbf{v})$ and $\mathbf{v}^k \in \partial\|\cdot\|_{(K)}(\mathbf{w}^k)$, we have

$$\begin{aligned} \iota_{S_{(K)}^*}(\mathbf{v}^k) - \langle \mathbf{w}^{k+1}, c\mathbf{v}^k \rangle &= \langle \mathbf{w}^k, c\mathbf{v}^k \rangle - c\|\cdot\|_{(K)}(\mathbf{w}^k) - \langle \mathbf{w}^{k+1}, c\mathbf{v}^k \rangle \\ &= -c\|\cdot\|_{(K)}(\mathbf{w}^k) - \langle \mathbf{w}^{k+1} - \mathbf{w}^k, c\mathbf{v}^k \rangle, \end{aligned} \tag{41}$$

where the first equality is from the Young's inequality. Summing (40) and (41) leads to

$$H(\mathbf{w}^{k+1}, \mathbf{v}^k) \leq Q(\mathbf{w}^k) - \frac{1}{2\alpha}\|\mathbf{w}^{k+1} - \mathbf{w}^k\|_2^2.$$

Finally, this together with (19) implies Item (i). We complete the proof immediately. □

## A.10. Proof of Proposition 3.11

We require to review the notion of KL property. This property and the associated notion of KL exponent have been used extensively in the convergence analysis of various first-order methods (Attouch & Bolte, 2009; Attouch et al., 2010; 2013; Bolte et al., 2014).

**Definition A.1.** (Kurdyka-Lojasiewicz (KL) property and exponent) We say that a proper function $\varphi : \mathbb{R}^n \to (-\infty, +\infty]$ satisfies the Kurdyka-Lojasiewicz (KL) property at an $\widehat{x} \in \mathrm{dom}\,\partial\varphi$ if there are $a \in (0, +\infty]$, a neighborhood $U$ of $\widehat{x}$, and a continuous concave function $\phi : [0, a) \to [0, +\infty)$ with $\phi(0) = 0$ such that:

(i) $\phi$ is continuously differentiable on $(0, a)$ with $\phi' > 0$ on $(0, a)$;

(ii) for every $x \in U$ with $\varphi(\widehat{x}) < \varphi(x) < \varphi(\widehat{x}) + a$, it holds that

$$\phi'(\varphi(x) - \varphi(\widehat{x}))\,\mathrm{dist}(0, \partial\varphi(x)) \geq 1. \tag{42}$$

If $\varphi$ satisfies the KL property at $\widehat{x} \in \mathrm{dom}\,\partial\varphi$ and the $\phi$ in (42) can be chosen as $\phi(\mathbf{v}) = a_0\mathbf{v}^{1-\theta}$ for some $a_0 > 0$ and $\theta \in [0, 1)$, then we say that $\varphi$ satisfies the KL property at $\widehat{x}$ with exponent $\theta$.

A proper function $\varphi : \mathbb{R}^n \to (-\infty, +\infty]$ is called a KL function if it satisfies the KL property at each point of $\mathrm{dom}\,\partial\varphi$. KL functions encompass extensive functions arisen in various applications. For instance, a subanalytic function ( Facchinei and Pang, 2003, Definition 6.6.1) $\varphi$, which is continuous around $x \in \mathrm{dom}\,\partial\varphi$, satisfies the KL property at $x$ (Bolte et al., 2014, Theorem 3.1). Moreover, a proper semi-algebraic function $\varphi$, which is lower semicontinuous around $x \in \mathrm{dom}\,\partial\varphi$, satisfies the KL property at $x$ with some exponent $\theta \in [0, 1)$ (Attouch et al., 2010; Bolte et al., 2014).

We next review the notion of semi-algebraic function. A function $\varphi : \mathbb{R}^n \to (-\infty, +\infty]$ is said to be semialgebraic if its graph $\mathrm{Graph}(\varphi) := \{(\mathbf{x}, s) \in \mathbb{R}^n \times \mathbb{R} : s = \varphi(x)\}$ is a semialgebraic subset of $\mathbb{R}^{n+1}$, that is, there exist a finite number of real polynomial functions $G_{ij}, H_{ij} : \mathbb{R}^{n+1} \to \mathbb{R}$ such that

$$\mathrm{Graph}(\varphi) = \bigcup_{j=1}^{p} \bigcap_{i=1}^{q} \{\mathbf{z} \in \mathbb{R}^{n+1} : G_{ij}(\mathbf{z}) = 0, H_{ij}(\mathbf{z}) < 0\}.$$

It is obvious that $H$ is a lower semicontinuous function due to the continuity of $f$ and the closedness of $\Delta_m$ and $S^*_{(K)}$. By simple calculation, we have

$$
\begin{aligned}
\mathrm{Graph}(H) =& \{(\mathbf{w}, \mathbf{v}, z) \in \mathbb{R}^m \times \mathbb{R}^m \times \mathbb{R} : z = H(\mathbf{w}, \mathbf{v})\} \\
=& \{(\mathbf{w}, \mathbf{v}, z) \in \mathbb{R}^m \times \mathbb{R}^m \times \mathbb{R} : -\mathbf{q}_{[t]}^\top \mathbf{w} + \lambda \|\mathbf{w} - \hat{\mathbf{w}}_{[t]}\|_1 - \langle \mathbf{w}, c\mathbf{v} \rangle = z, \ \mathbf{w} \geq 0, \\
& \sum_{i=1}^m \mathbf{w}_i = 1, \ \|\mathbf{v}\|_\infty \leq 1, \ \|\mathbf{v}\|_1 \leq K \} \\
=& \bigcup_{a_i \in \{-1,1\}} \bigcap_{d_i \in \{-1,1\}} \{(\mathbf{w}, \mathbf{v}, z) \in \mathbb{R}^m \times \mathbb{R}^m \times \mathbb{R} : -\mathbf{q}_{[t]}^\top \mathbf{w} + \lambda \sum_{i=1}^m a_i (\mathbf{w} - \hat{\mathbf{w}}_{[t]})_i - \\
& \sum_{i=1}^m c\mathbf{w}_i \mathbf{v}_i - z = 0, \ -a_i(\mathbf{w} - \hat{\mathbf{w}}_{[t]})_i \leq 0, \ -\mathbf{w}_i \leq 0, \ \mathbf{v}_i - 1 \leq 0 \text{ for } i = 1, 2, \ldots, m, \\
& \langle \mathbf{w}, \mathbf{1}_m \rangle = 1, \ \sum_{i=1}^m \mathbf{v}_i d_i - K \leq 0 \}.
\end{aligned}
$$

This implies that $H$ is a semialgebraic function. Therefore, $H$ is a KL function due to the lower semicontinuity of $H$.

We next show the KL exponent of $H$ is $1/2$. In view of the definition of $H$, we can rewrite it as

$$
H(\mathbf{w}, \mathbf{v}) = \min_{\mathbf{u} \in \Gamma} \{Q_u(\mathbf{x}) + P_u(\mathbf{x})\},
$$

where $\Gamma := \{\mathbf{u} \in \mathbb{R}^m : \mathbf{u}_i \in \{-1, 1\}, \forall i\}$, $Q_u$ are quadratic functions and $P_u$ are polyhedral functions indexed by $\mathbf{u}$. Specifically, for each $\mathbf{u} \in \Gamma$, one can define $P_u$ as the indicator function on the set $\{(\mathbf{w}, \mathbf{v}) : \mathbf{w} \in \Delta_m, \|\mathbf{v}\|_\infty \leq 1, \|\mathbf{v}\|_1 \leq K, \mathbf{u} \circ (\mathbf{w} - \hat{\mathbf{w}}_{[t]}) \geq 0\}$, and $Q_u$ is chosen as $-\mathbf{q}_{[t]}^\top \mathbf{w} + \lambda \langle \mathbf{u}, \mathbf{w} - \hat{\mathbf{w}}_{[t]} \rangle - \langle \mathbf{w}, c\mathbf{v} \rangle$, where $\circ$ is the Hadamard product. It is obvious that $P_u$ is a polyhedral function and $Q_u$ is a quadratic function for each $\mathbf{u} \in \Gamma$. Then, invoking (Bolte et al., 2007, Corollary 5.2), $H$ satisfies the KL property at each point of $\Delta_m \times S^*_{(K)}$ with exponent $1/2$.

### A.11. Proof of Theorem 3.12

We first review a framework for proving sequential convergence using the KL property.

**Proposition A.2.** *(Li et al., 2022, Proposition 2.7) Let $\Psi : \mathbb{R}^n \times \mathbb{R}^m \to (-\infty, +\infty]$ be proper lower semicontinuous. Consider a bounded sequence $\{(\mathbf{u}^k, \mathbf{v}^k) \in \mathbb{R}^n \times \mathbb{R}^m : k \in \mathbb{N}\}$ satisfying the following three conditions:*

*(i) (Sufficient descent condition). There exists $a > 0$ such that*

$$
\Psi(\mathbf{u}^{k+1}, \mathbf{v}^{k+1}) + a\|\mathbf{u}^{k+1} - \mathbf{u}^k\|_2^2 \leq \Psi(\mathbf{u}^k, \mathbf{v}^k), \quad \text{for all } k \in \mathbb{N};
$$

*(ii) (Relative error condition). There exists $b > 0$, such that*

$$
\mathrm{dist}(0, \partial\Psi(\mathbf{u}^{k+1}, \mathbf{v}^{k+1})) \leq b\|\mathbf{u}^{k+1} - \mathbf{u}^k\|_2, \quad \text{for all } k \in \mathbb{N};
$$

*(iii) (Continuity condition). The limit $\Psi_\infty := \lim_{k\to\infty} \Psi(\mathbf{u}^k, \mathbf{v}^k)$ exists and $\Psi \equiv \Psi_\infty$ holds on $\Upsilon$, where $\Upsilon$ denotes the set of accumulation points of $\{(\mathbf{u}^k, \mathbf{v}^k) : k \in \mathbb{N}\}$.*

*If $\Psi$ satisfies the KL property at each point of $\Upsilon$, then we have $\sum_{k=0}^\infty \|\mathbf{u}^{k+1} - \mathbf{u}^k\|_2 < +\infty$, $\lim_{k\to\infty} \mathbf{u}^k = \mathbf{u}^*$ and $0 \in \partial\Psi(\mathbf{u}^*, \mathbf{v}^*)$ for some $(\mathbf{u}^*, \mathbf{v}^*) \in \Upsilon$.*

Next, we present a proposition regarding the relative error condition for the sequence generated by PSGA.

**Proposition A.3.** *Let $\{(\mathbf{w}^k, \mathbf{v}^k) : k \in \mathbb{N}\}$ be generated by PSGA for any arbitrary $\mathbf{w}^0 \in \Delta_m$. Then there exists $b > 0$ such that*

$$
\mathrm{dist}(0, \partial H(\mathbf{w}^{k+1}, \mathbf{v}^k)) \leq b\|\mathbf{w}^{k+1} - \mathbf{w}^k\|_2. \tag{43}
$$

*Proof.* It is obvious that

$$\partial H(\mathbf{w}^{k+1}, \mathbf{v}^k) = \left(\partial f(\mathbf{w}^{k+1}) + \partial \iota_{\Delta_m}(\mathbf{w}^{k+1}) - c\mathbf{v}\right) \times \left(\partial \iota_{S^*_{(K)}}(\mathbf{v}^k) - c\mathbf{w}^{k+1}\right).$$

In order to show (43), it suffices to prove

$$\text{dist}(0, \partial f(\mathbf{w}^{k+1}) + \partial \iota_{\Delta_m}(\mathbf{w}^{k+1}) - c\mathbf{v}) \le b_1 \|\mathbf{w}^{k+1} - \mathbf{w}^k\|_2 \tag{44}$$

and

$$\text{dist}(0, \partial \iota_{S^*_{(K)}}(\mathbf{v}^k) - c\mathbf{w}^{k+1}) \le b_2 \|\mathbf{w}^{k+1} - \mathbf{w}^k\|_2 \quad \text{for some } b_1, b_2 > 0. \tag{45}$$

Since $\mathbf{v}^k \in \partial \|\cdot\|_{(K)}(\mathbf{w}^k)$ from PSGA, we have $\mathbf{w}^k \in \partial\left(c\|\cdot\|_{(K)}\right)^*(c\mathbf{v}^k)$. Invoking that $\left(c\|\cdot\|_{(K)}\right)^*(\mathbf{v}) = \iota_{S^*_{(K)}}(\mathbf{v})$, this leads to

$$c\mathbf{w}^k \in \partial \iota_{S^*_{(K)}}(\mathbf{v}^k).$$

Therefore,

$$\text{dist}\left(0, \partial \iota_{S^*_{(K)}}(\mathbf{v}^k) - c\mathbf{w}^{k+1}\right) = \text{dist}\left(0, \partial \iota_{S^*_{(K)}}(\mathbf{v}^k) - c\mathbf{w}^k + c\mathbf{w}^k - c\mathbf{w}^{k+1}\right)$$
$$\le c\|\mathbf{w}^{k+1} - \mathbf{w}^k\|_2,$$

which implies (45). We next prove (44).

According to (11), we deduce from the definition of proximity operators and the generalized Fermat's Rule that

$$\mathbf{w}^k + \alpha c\mathbf{v}^k - \mathbf{w}^{k+1} \in \alpha \partial f(\mathbf{w}^{k+1}) + \partial \iota_{\Delta_m}(\mathbf{w}^{k+1}).$$

This yields that

$$\frac{\mathbf{w}^k - \mathbf{w}^{k+1}}{\alpha} \in \partial f(\mathbf{w}^{k+1}) + \partial \iota_{\Delta_m}(\mathbf{w}^{k+1}) - c\mathbf{v}^k$$

due to $\alpha \iota_{\Delta_m} = \iota_{\Delta_m}$ for any $\alpha > 0$. Then, (44) follows immediately. We complete the proof. $\square$

Now, we are ready to show Theorem 3.12.

*Proof.* Since $\mathbf{w}^k \in \Delta_m$ for $k \in \mathbb{N}$ and $\mathbf{v}^k \in \partial \|\cdot\|_{(K)}(\mathbf{w}^k)$, we have that $\{(\mathbf{w}^k, \mathbf{v}^k) : k \in \mathbb{N}\}$ is bounded due to the convexity of $\|\cdot\|_{(K)}$. Then, Item (i) follows from Proposition A.2, Proposition A.3, Proposition 3.11, and the continuity of $f$. Items (ii) and (iii) are direct results of Proposition 3.5. $\square$

## A.12. Proof of Theorem 3.13

In order to show the $R$-linear convergence rate of PSGA, we first establish a proposition regarding the general convergence rate of PSGA under the KL exponent assumption. The proof is very similar to those for other first-order algorithms based on the KL exponent; see, for example, (Attouch & Bolte, 2009; Wright et al., 2009; Yin et al., 2014) . Therefore, we omit the detail of the proof here.

**Proposition A.4.** *Let $\{\mathbf{w}^k : k \in \mathbb{N}\}$ be generated by PSGA for any arbitrary $\mathbf{w}^0 \in \Delta_m$ and suppose that $\{\mathbf{w}^k : k \in \mathbb{N}\}$ converges to $\mathbf{w}^*$. Assume that $H$ satisfies the KL property with exponent $\theta \in [0, 1)$. Then, the following statements hold:*

*(i) If $\theta = 0$, then $\{\mathbf{w}^k : k \in \mathbb{N}\}$ converges to $\mathbf{w}^*$ finitely;*

*(ii) If $\theta \in (0, \frac{1}{2}]$, then $\|\mathbf{w}^k - \mathbf{w}^*\|_2 \le a_0 b_0^k$ for any $k \ge k_0$ with $k_0 > 0$, $a_0 > 0$, $b_0 \in (0, 1)$;*

*(iii) If $\theta \in (\frac{1}{2}, 1)$, then $\|\mathbf{w}^k - \mathbf{w}^*\|_2 \le a_1 k^{-(1-\theta)/(2\theta-1)}$ for any $k \ge k_1$ with some $k_1 > 0$ and $a_1 > 0$.*

Then, Theorem 3.13 is a direct consequence of Theorem 3.12, Proposition A.4 and Proposition 3.11.

# B. Experimental Results

## B.1. Parameter Sensitivity

We evaluate the sensitivity of parameters $K$ and $c$ to illustrate the reasonableness of the selected parameter values and the robustness of the proposed STCO method. We first examine the effect of the sparsity parameter $K$ by conducting experiments for $K = 10$ and $K = 15$. The cumulative wealth results of STCO1 and STCO2 under sparsity levels $K = 10$ and $K = 15$ in TSE and MSCI datasets are presented in Table 2. It can be observed that the cumulative wealth is almost unchanged when $K$ varies from 10 to 15, indicating that the proposed method is insensitive to $K$ within a moderate range. This also supports the choice of $K = 10$ used in our experiments.

*Table 2.* Cumulative wealth of STCO1 and STCO2 under sparsity levels $K = 10$ and $K = 15$ in TSE and MSCI datasets.

| ALGORITHMS ($\gamma = 0\%$) | TSE | | MSCI | |
| --- | --- | --- | --- | --- |
| | K=10 | K=15 | K=10 | K=15 |
| STCO1 | 1216.92 | 1216.92 | 11.48 | 11.48 |
| STCO2 | 95.92 | 95.92 | 15.17 | 15.17 |

| ALGORITHMS ($\gamma = 0.25\%$) | TSE | | MSCI | |
| --- | --- | --- | --- | --- |
| | K=10 | K=15 | K=10 | K=15 |
| STCO1 | 5.60 | 5.60 | 2.65 | 2.65 |
| STCO2 | 3.97 | 3.97 | 2.05 | 2.05 |

Moreover, we provide a detailed discussion for the practical choice of the penalty parameter $c$ without exhaustive tuning. The condition $c > 2\sqrt{2}L$ in Theorem 3.3 provides a sufficient theoretical guideline for choosing $c$. In practice, however, $L$ may be difficult to estimate accurately and we observed that the proposed method is not highly sensitive to the exact value of $c$ as long as it lies within a reasonable range. As shown in Table 3, the cumulative wealth remains very stable across different values of $c$. For example, on NYSE(N) with $\gamma = 0.25\%$, the cumulative wealth of STCO1 remains around 274.18, while that of STCO2 remains around 6.00E+5; on TSE with $\gamma = 0\%$, the cumulative wealth of STCO1 remains around 1216.92, and that of STCO2 remains around 95.92. These results indicate that the portfolio performance is not highly sensitive to $c$ within a moderate range. Therefore, we set $c = 8 \times 10^{-6}$ in all experiments.

*Table 3.* Cumulative wealth of STCO1 and STCO2 under different values of $c$ on the NYSE(N) dataset with $\gamma = 0.25\%$ and on the TSE dataset with $\gamma = 0\%$.

| $c$ VALUES | NYSE(N) ($\gamma = 0.25\%$) | | TSE ($\gamma = 0\%$) | |
| --- | --- | --- | --- | --- |
| | STCO1 | STCO2 | STCO1 | STCO2 |
| 2E-6 | 274.15 | 6.00E+5 | 1216.92 | 95.92 |
| 4E-6 | 274.18 | 6.00E+5 | 1216.92 | 95.92 |
| 8E-6 | 274.18 | 6.00E+5 | 1216.92 | 95.92 |
| 1.6E-5 | 274.18 | 6.00E+5 | 1216.92 | 95.92 |

## B.2. Datasets Information

We conduct numerical experiments on 4 benchmark datasets, which are NYSE(O), NYSE(N), TSE and MSCI. They collects historical relative price from New York Exchange, Toronto Stock Exchange and the stock markets of 24 countries around the world, where the element in $i$-th row and $j$-th column represents the daily relative price of $j$-th asset during $i$-th period. Detail is shown in Table 4.

## B.3. Comparison Approaches

All compared strategies and their parameter value are detailed below:

a) Ubah: Uniformly buy-and-hold strategy;

b) Best: Best-stock in hindsight;

*Table 4.* Information of four benchmark data-sets from real markets.

| Datasets | Region | Time | Days | Stocks |
|----------|--------|------|------|--------|
| NYSE(O) | US | 07/$Mar$/1962 $\smile$ 31/$Dec$/1984 | 5651 | 36 |
| NYSE(N) | US | 01/$Jan$/1985 $\smile$ 30/$Jun$/2010 | 6431 | 23 |
| TSE | Toronto | 04/$Jan$/1994 $\smile$ 31/$Dec$/1998 | 1259 | 88 |
| MSCI | 24 countries | 01/$Apr$/2006 $\smile$ 31/$Mar$/2010 | 1043 | 24 |

c) Bcrp : Best Constant Rebalanced Portfolios in hindsight;

d) TCO1 : Transaction Cost Optimization, parameters are set to $\lambda = 10\gamma$, $\eta = 10$;

e) TCO2 : Transaction Cost Optimization, parameters are set to $\lambda = 10\gamma$, $w = 4$, $\eta = 10$;

f) SSPO : Short-term Sparse Portfolio Optimization, parameters are set to $\eta = 0.005$, $\zeta = 500$, $w = 5$, $\lambda = 0.5$, $\gamma = 0.01$;

g) S1 : parameters are set to $w = 5$, $\epsilon = 0.001$ (NYSE(O), NYSE(N)), $\epsilon = 0.01$ (TSE), $\epsilon = 0.003$ (MSCI);

h) S2 : parameters are set to $w = 5$, $\epsilon = 0.001$ (NYSE(O), NYSE(N)), $\epsilon = 0.01$ (TSE), $\epsilon = 0.003$ (MSCI);

i) S3 : parameters are set to $w = 5$, $s_t = 3$, $\epsilon = 0.001$ (NYSE(O), NYSE(N)), $\epsilon = 0.01$ (TSE), $\epsilon = 0.003$ (MSCI).

j) DENRPO1-PARM : parameters are set to $\alpha = \frac{0.999}{\rho m}$, $\rho = 0.618$, $\lambda = 10\gamma$, $\eta = 0.00025$ and $\tau = 0.00005$.

k) DENRPO1-OLMAR : parameters are set to $\alpha = \frac{0.999}{\rho m}$, $\rho = 0.618$, $\lambda = 10\gamma$, $\eta = 0.00025$ and $\tau = 0.00005$.

l) DENRPO2-PARM : parameters are set to $\alpha = \frac{0.999}{\rho m}$, $\rho = 0.618$, $\lambda = 10\gamma$, $\eta = 0.00025$ and $\tau = 0.00005$.

m) DENRPO2-OLMAR : parameters are set to $\alpha = \frac{0.999}{\rho m}$, $\rho = 0.618$, $\lambda = 10\gamma$, $\eta = 0.00025$ and $\tau = 0.00005$.

The codes of PSGA-ADMM are accessible via the link: `https://github.com/Ting221/STCO`.

## B.4. Calculus Rules of Return and Risk Factors

Different from the normal MER, we develop to represent the net proportion gained or lost wealth with transaction cost as follows:

$$r_t = (\mathbf{w}_{[t]}^\top \mathbf{x}_{[t]} - 1) * \theta_{[t-1]}.$$

Hence, MER can be denoted by:

$$MER = \bar{r}_s - \bar{r}_m = \frac{1}{T} \sum_{t=1}^{T} r_{s,t} - r_{m,t},$$

where $r_{s,t}$ and $r_{m,t}$ denote daily excess returns of a portfolio strategy and Ubah benchmark on the $t$-th day, respectively.

Moreover, Sharpe (1964) shows that the $\alpha$ factor can be expressed as follows:

$$\hat{\beta} = \frac{\hat{c}(r_s, r_m)}{\hat{\sigma}^2(r_m)}, \tag{46}$$

$$\hat{\alpha} = \bar{r}_s - \hat{\beta}\bar{r_m}, \tag{47}$$

where $\hat{c}(\cdot, \cdot)$ and $\hat{\sigma}(\cdot)$ denotes the sample covariance and the standard deviation computed over a period of T trading days, respectively. Besides, $\beta$ factor can be computed as shown in (46).

## B.5. Results for Cumulative Wealth, Return and Risk Factors

Experimental results of Cumulative Wealth with transaction cost rate 0%, 0.25% and 0.5% are presented in Table 5, meanwhile MER, $\alpha$ Factor and $\beta$ Factor are detailed in Table 6.

*Table 5.* Cumulative wealth with transaction costs from the proposed STCO1 and STCO2 and the compared algorithms in four data-sets, where UBAH, BEST and BCRP are benchmarks, SSPO, S1, S2 and S3 are strategies without considering transaction costs, and TCOs and DENRPOs are transaction cost optimization algorithms. Besides, we highlight the top two values in each column.

| ALGORITHMS | NYSE(O) | | | NYSE(N) | | |
|---|---|---|---|---|---|---|
| | 0% | 0.25% | 0.5% | 0% | 0.25% | 0.5% |
| UBAH | 14.50 | 14.46 | 14.43 | 18.06 | 18.01 | 17.97 |
| BEST | 54.14 | 54.01 | 53.87 | 83.51 | 83.30 | 83.09 |
| BCRP | 252.07 | 181.03 | 133.52 | 119.71 | 98.02 | 80.23 |
| SSPO | 1.06E+18 | **2.45E+11** | 5.66E+04 | 1.62E+09 | 154.92 | 0.00 |
| S1 | **1.24E+18** | 8.27E+10 | 5.53E+03 | **2.97E+09** | 93.65 | 0.00 |
| S2 | **1.29E+18** | 8.12E+10 | 5.62E+03 | 2.87E+09 | 93.95 | 0.00 |
| S3 | 1.24E+18 | 8.26E+10 | 5.52E+03 | **2.97E+09** | 93.46 | 0.00 |
| TCO1 | 1.35E+14 | 5.57E+09 | 2.33E+06 | 9.15E+06 | **3.81E+03** | 143.47 |
| TCO2 | 1.47E+13 | 4.34E+07 | 1.52E+04 | 2.35E+07 | 2.14E+03 | 57.61 |
| DENRPO1-PARM | 5.78E+15 | 2.10E+11 | **7.99E+06** | 9.61E+05 | 296.58 | 55.02 |
| DENRPO1-OLMAR | 3.57E+16 | 7.36E+09 | 6.59E+04 | 2.19E+08 | 2.92E+03 | **893.22** |
| DENRPO2-PARM | 5.78E+15 | 2.10E+11 | **8.02E+06** | 9.61E+05 | 297.64 | 54.27 |
| DENRPO2-OLMAR | 3.57E+16 | 7.37E+09 | 6.61E+04 | 2.19E+08 | 2.91E+03 | **886.46** |
| **STCO1** | 6.16E+15 | **3.04E+11** | 6.54E+06 | 7.48E+05 | 274.18 | 53.88 |
| **STCO2** | 7.59E+16 | 1.24E+10 | 1.07E+05 | 3.46E+08 | **6.00E+05** | 271.82 |

| ALGORITHMS | TSE | | | MSCI | | |
|---|---|---|---|---|---|---|
| | 0% | 0.25% | 0.5% | 0% | 0.25% | 0.5% |
| UBAH | 1.61 | 1.61 | 1.60 | 0.91 | 0.90 | 0.90 |
| BEST | 6.28 | 6.26 | 6.25 | 1.50 | 1.50 | 1.50 |
| BCRP | 6.70 | 6.38 | 6.12 | 1.51 | 1.49 | 1.48 |
| SSPO | 364.94 | **11.78** | 0.38 | 7.51 | 0.38 | 0.02 |
| S1 | 227.29 | 6.73 | 0.20 | 10.17 | 0.43 | 0.02 |
| S2 | 249.08 | 6.18 | 0.15 | 10.35 | 0.39 | 0.01 |
| S3 | 227.66 | 6.76 | 0.20 | 10.16 | 0.42 | 0.02 |
| TCO1 | 149 | 7.66 | 0.91 | 9.68 | 1.52 | 1.13 |
| TCO2 | 152.98 | **31.71** | **4.99** | 5.66 | 1.42 | 0.84 |
| DENRPO1-PARM | **1471.29** | 6.89 | 1.83 | 10.84 | **2.62** | **1.30** |
| DENRPO1-OLMAR | 365.16 | 3.93 | 1.53 | **16.21** | 2.15 | 0.96 |
| DENRPO2-PARM | **1471.12** | 6.89 | 1.84 | 10.61 | 2.48 | 1.21 |
| DENRPO2-OLMAR | 365.18 | 3.93 | 1.52 | **16.21** | 2.15 | 0.96 |
| **STCO1** | 1216.92 | 5.60 | 1.49 | 11.48 | **2.65** | **1.21** |
| **STCO2** | 95.92 | 3.97 | **4.88** | 15.17 | 2.05 | 0.76 |

## B.6. Results for Sparsity

Experimental results of the average and standard deviation of sparsity of the portfolios generated by STCO1 and STCO2 and other state-of-the-art algorithms with $K = 10$ under transaction cost rate $\gamma = 0\%$ and $0.25\%$ are summarized in Table 7. The results show that the proposed STCO methods select only a small number of assets on average across all datasets and transaction cost settings, which confirms the sparsity-promoting effect of the proposed model.

*Table 6.* MER, $\alpha$ Factor and $\beta$ Factor achieved by the proposed STCO1 and STCO2 algorithms, in comparison with TCOs and DENRPOs on four datasets with transaction costs. In order to emphasize the results under transaction costs, we only present the comparison with transaction costs optimization algorithms, which are TCOs and DENRPOs. Additionally, the top two performances in each column (excluding BEST benchmark rows) are highlighted in bold.

| ALGORITHMS (**MER**) | NYSE(O) 0% | 0.25% | 0.5% | NYSE(N) 0% | 0.25% | 0.5% | TSE 0% | 0.25% | 0.5% | MSCI 0% | 0.25% | 0.5% |
|---|---|---|---|---|---|---|---|---|---|---|---|---|
| BEST | 0.0003 | 0.0003 | 0.0003 | 0.0003 | 0.0003 | 0.0003 | 0.0016 | 0.0016 | 0.0016 | 0.0004 | 0.0004 | 0.0004 |
| TCO1 | 0.0056 | 0.0051 | 0.0035 | 0.0024 | 0.0020 | 0.0013 | 0.0047 | 0.0046 | 0.0028 | 0.0024 | 0.0009 | 0.0003 |
| TCO2 | 0.0052 | 0.0038 | 0.0020 | 0.0026 | 0.0017 | 0.0010 | 0.0048 | 0.0050 | 0.0034 | 0.0019 | 0.0007 | 0.0001 |
| DENRPO1-PARM | 0.0065 | 0.0069 | **0.0038** | 0.0022 | 0.0023 | 0.0014 | **0.0068** | **0.0060** | **0.0039** | 0.0026 | 0.0017 | **0.0005** |
| DENRPO1-OLMAR | 0.0069 | 0.0052 | 0.0024 | 0.0031 | 0.0021 | **0.0015** | 0.0059 | 0.0040 | 0.0027 | 0.0029 | 0.0012 | 0.0001 |
| DENRPO2-PARM | 0.0065 | **0.0069** | **0.0038** | 0.0022 | **0.0023** | 0.0014 | **0.0068** | **0.0060** | **0.0039** | 0.0025 | **0.0017** | **0.0004** |
| DENRPO2-OLMAR | 0.0069 | 0.0052 | 0.0024 | **0.0031** | 0.0021 | **0.0015** | 0.0059 | 0.0040 | 0.0027 | **0.0029** | 0.0012 | 0.0001 |
| STCO1 | 0.0065 | **0.0070** | 0.0037 | 0.0021 | 0.0022 | 0.0014 | 0.0065 | 0.0058 | **0.0039** | 0.0026 | **0.0018** | **0.0004** |
| STCO2 | **0.0070** | 0.0053 | 0.0026 | **0.0032** | **0.0030** | 0.0014 | 0.0049 | 0.0039 | 0.0037 | **0.0029** | 0.0012 | 0.0000 |

| ALGORITHMS ($\alpha$ **FACTOR**) | NYSE(O) 0% | 0.25% | 0.5% | NYSE(N) 0% | 0.25% | 0.5% | TSE 0% | 0.25% | 0.5% | MSCI 0% | 0.25% | 0.5% |
|---|---|---|---|---|---|---|---|---|---|---|---|---|
| BEST | 0.0003 | 0.0003 | 0.0003 | 0.0003 | 0.0004 | 0.0004 | 0.0015 | 0.0015 | 0.0015 | 0.0005 | 0.0005 | 0.0005 |
| TCO1 | 0.0055 | 0.0050 | 0.0034 | 0.0023 | 0.0020 | 0.0013 | 0.0045 | 0.0044 | 0.0026 | 0.0024 | 0.0009 | 0.0003 |
| TCO2 | 0.0051 | 0.0036 | 0.0019 | 0.0025 | 0.0017 | 0.0010 | 0.0046 | 0.0048 | 0.0032 | 0.0019 | 0.0007 | 0.0000 |
| DENRPO1-PARM | 0.0064 | **0.0067** | **0.0038** | 0.0021 | 0.0022 | 0.0014 | **0.0067** | 0.0056 | **0.0038** | 0.0026 | **0.0017** | **0.0005** |
| DENRPO1-OLMAR | 0.0067 | 0.0050 | 0.0024 | 0.0030 | 0.0020 | **0.0015** | 0.0057 | 0.0038 | 0.0026 | 0.0029 | 0.0012 | 0.0001 |
| DENRPO2-PARM | 0.0064 | 0.0067 | **0.0038** | 0.0021 | 0.0022 | 0.0014 | **0.0067** | **0.0057** | **0.0038** | 0.0025 | 0.0017 | 0.0004 |
| DENRPO2-OLMAR | **0.0067** | 0.0050 | 0.0024 | **0.0030** | 0.0020 | **0.0015** | 0.0057 | 0.0038 | 0.0026 | **0.0029** | 0.0012 | 0.0001 |
| STCO1 | 0.0064 | **0.0068** | 0.0037 | 0.0021 | **0.0022** | 0.0014 | 0.0064 | **0.0056** | 0.0036 | 0.0026 | **0.0017** | **0.0005** |
| STCO2 | **0.0069** | 0.0051 | 0.0025 | **0.0031** | **0.0030** | 0.0013 | 0.0046 | 0.0036 | 0.0034 | **0.0029** | 0.0012 | 0.0000 |

| ALGORITHMS ($\beta$ **FACTOR**) | NYSE(O) 0% | 0.25% | 0.5% | NYSE(N) 0% | 0.25% | 0.5% | TSE 0% | 0.25% | 0.5% | MSCI 0% | 0.25% | 0.5% |
|---|---|---|---|---|---|---|---|---|---|---|---|---|
| BEST | 0.9169 | 0.9169 | 0.9169 | 0.8904 | 0.8904 | 0.8904 | 1.4552 | 1.4551 | 1.4551 | 0.4585 | 0.4585 | 0.4585 |
| TCO1 | **1.2092** | **1.2428** | 1.1515 | **1.0799** | **1.0398** | 0.9972 | 1.4721 | **1.5109** | 1.5131 | **1.1279** | **1.0780** | **0.9270** |
| TCO2 | 1.2781 | 1.3211 | 1.1964 | 1.1450 | 1.0935 | 1.0614 | 1.5308 | 1.5476 | 1.5499 | 1.1532 | 1.1601 | 1.1264 |
| DENRPO1-PARM | 1.2354 | 1.2474 | **1.0851** | 1.0855 | 1.0492 | **0.9956** | **1.3099** | 1.5395 | **1.2146** | 1.1289 | 1.1202 | **0.9684** |
| DENRPO1-OLMAR | 1.3124 | 1.3454 | 1.1309 | 1.1817 | 1.1848 | 1.0607 | 1.5291 | 1.5904 | 1.4146 | 1.1789 | 1.1297 | 1.0289 |
| DENRPO2-PARM | 1.2354 | **1.2474** | 1.0852 | 1.0855 | **1.0492** | **0.9954** | 1.3099 | 1.5395 | 1.2146 | 1.1270 | 1.1231 | 0.9750 |
| DENRPO2-OLMAR | 1.3125 | 1.3454 | 1.1310 | 1.1817 | 1.1849 | 1.0602 | 1.5291 | 1.5904 | 1.4139 | 1.1789 | 1.1297 | 1.0289 |
| STCO1 | **1.2288** | 1.2533 | **1.0839** | **1.0840** | 1.0504 | 0.9968 | **1.3058** | **1.5341** | **1.2108** | **1.1262** | 1.1219 | 0.9736 |
| STCO2 | 1.2774 | 1.3226 | 1.1812 | 1.1461 | 1.1145 | 1.1114 | 1.6606 | 1.6350 | 1.6758 | 1.1782 | **1.0570** | 1.1675 |

*Table 7.* Average and standard deviation of sparsity of the portfolios generated by STCO1 and STCO2 and other state-of-the-art algorithms under transaction cost rate $\gamma = 0\%$ and $0.25\%$

| ALGORITHM | STATISTIC | NYSE(O) | | NYSE(N) | | TSE | | MSCI | |
|---|---|---|---|---|---|---|---|---|---|
| | | $\gamma = 0\%$ | $\gamma = 0.25\%$ | $\gamma = 0\%$ | $\gamma = 0.25\%$ | $\gamma = 0\%$ | $\gamma = 0.25\%$ | $\gamma = 0\%$ | $\gamma = 0.25\%$ |
| S1 | MEAN | 1.0488 | 1.0488 | 1.0706 | 1.0706 | 1.0921 | 1.0921 | 1.2061 | 1.2061 |
| | STD | 0.5119 | 0.5119 | 0.6573 | 0.6573 | 2.4562 | 2.4562 | 1.4651 | 1.4651 |
| S2 | MEAN | 1.0062 | 1.0062 | 1.0034 | 1.0034 | 1.0691 | 1.0691 | 1.0221 | 1.0221 |
| | STD | 0.4656 | 0.4656 | 0.2743 | 0.2743 | 2.4519 | 2.4519 | 0.7122 | 0.7122 |
| S3 | MEAN | 1.0487 | 1.0487 | 1.0580 | 1.0580 | 1.0921 | 1.0921 | 1.1438 | 1.1438 |
| | STD | 0.5110 | 0.5110 | 0.3664 | 0.3664 | 2.4562 | 2.4562 | 0.8045 | 0.8045 |
| TCO1 | MEAN | 7.6556 | 7.5861 | 7.5116 | 7.4886 | 6.7665 | 6.7482 | 10.4688 | 10.6616 |
| | STD | 3.5488 | 3.5078 | 3.2425 | 3.2964 | 4.1446 | 4.1448 | 3.8044 | 4.1774 |
| TCO2 | MEAN | 6.2674 | 6.1996 | 5.9793 | 5.9364 | 5.5941 | 5.5909 | 8.2042 | 8.1668 |
| | STD | 3.0564 | 3.0300 | 2.9270 | 2.9246 | 3.9788 | 3.9549 | 3.4682 | 3.7179 |
| SSPO | MEAN | 1.0143 | 1.0143 | 1.0089 | 1.0089 | 1.0778 | 1.0778 | 1.0297 | 1.0297 |
| | STD | 0.4741 | 0.4741 | 0.2840 | 0.2840 | 2.4534 | 2.4534 | 0.7173 | 0.7173 |
| DENRPO1-PARM | MEAN | 1.0322 | 1.0448 | 1.0361 | 1.1406 | 1.0818 | 1.1160 | 1.2550 | 2.5216 |
| | STD | 0.5052 | 0.5152 | 0.3341 | 0.6991 | 2.4541 | 2.4940 | 2.0577 | 3.9176 |
| DENRPO1-OLMAR | MEAN | 1.0379 | 1.0848 | 1.0373 | 1.1246 | 1.0921 | 1.2836 | 1.0738 | 5.7948 |
| | STD | 0.4978 | 0.8616 | 0.3316 | 1.0319 | 2.4562 | 3.9862 | 0.7469 | 5.8775 |
| DENRPO2-PARM | MEAN | 1.0322 | 1.0444 | 1.0361 | 1.1306 | 1.0818 | 1.1160 | 1.2550 | 2.8217 |
| | STD | 0.5052 | 0.5149 | 0.3341 | 0.6618 | 2.4541 | 2.4940 | 2.0577 | 4.2794 |
| DENRPO2-OLMAR | MEAN | 1.0379 | 1.0846 | 1.0373 | 1.1235 | 1.0921 | 1.2828 | 1.0738 | 5.7948 |
| | STD | 0.4978 | 0.8615 | 0.3316 | 1.0128 | 2.4562 | 3.9862 | 0.7469 | 5.8775 |
| STCO1 | MEAN | 1.0531 | 1.0821 | 1.0620 | 1.1827 | 1.0961 | 1.1461 | 1.3126 | 2.8849 |
| | STD | 0.5248 | 0.5546 | 0.3721 | 0.7330 | 2.4565 | 2.4985 | 2.0675 | 4.2975 |
| STCO2 | MEAN | 1.0541 | 1.1308 | 1.0571 | 1.1549 | 1.1064 | 1.3193 | 1.1189 | 2.1381 |
| | STD | 0.5145 | 0.8857 | 0.3602 | 0.8471 | 2.4582 | 4.0208 | 0.7783 | 3.9186 |

