# OpenReview forum: "A Linearly Convergent Proximal Subgradient Algorithm for Sparse Portfolio Optimization with Transaction Cost"
_ICML.cc/2026/Conference — ICML 2026 regular_

### Official Review · Reviewer_rJQ5 · 2026-02-25

**Soundness:** 4
**Presentation:** 4
**Significance:** 4
**Originality:** 3
**Overall Recommendation:** 5
**Confidence:** 5

**Summary:**

This paper addresses the problem of short-term portfolio optimization (SPO) by proposing a model that simultaneously considers transaction costs and portfolio sparsity, the K-sparse Transaction Cost Optimization (STCO) model. To solve the resulting NP-hard problem, the authors reformulate the K-sparse constraint into a nonsmooth Difference of Convex (DC) optimization problem using an exact penalty method. They then develop a Proximal Subgradient Algorithm (PSGA) that exploits the Alternating Direction Method of Multipliers (ADMM) to compute proximity operators. Theoretically, the paper establishes the global convergence of PSGA using the Kurdyka-Lojasiewicz (KL) property and proves an R-linear convergence rate by demonstrating a KL exponent of $1/2$.

**Compliance With Llm Reviewing Policy:**

Affirmed.

**Final Justification:**

The authors have fully addressed my questions, thus I have raised the score to recommend an acceptance of this paper.

**Key Questions For Authors:**

See weaknesses.

**Limitations:**

See weaknesses.

**Strengths And Weaknesses:**

Strengths:

•	The use of the KL property to prove linear convergence is a sophisticated approach. The equivalence proof between the sparse TCO and the DC reformulation (Theorem 3.3) provides an appropriate foundation for the proposed algorithm.

•	The paper identifies a gap in existing research, the independent treatment of transaction costs and sparsity. Then it provides a reasonable motivation for why they should be integrated to improve real-world investment stability.

•	The experiments use diverse metrics including Cumulative Wealth, Mean Excess Return (MER), and risk-adjusted factors ($\alpha$ and $\beta$). The comparison includes representative baselines like TCO, SSPO, and DENRPO.

Weaknesses:

•	While PSGA-ADMM is shown to be linearly convergent, the paper lacks a wall-clock time comparison against the baselines. Moreover, it involves an inner ADMM loop to compute the proximity operator. Hence it is better to provide a wall-clock time comparison against the baselines to see how PSGA-ADMM balances between investing return and runtime.

•	Theorem 3.3 requires the penalty parameter $c$ to be larger than $2\sqrt{2}L$, where $L$ is the Lipschitz constant of the objective function. In practice, $L$ may be difficult to estimate accurately, thus it is better to provide a detailed discussion on how to robustly select $c$ without exhaustive tuning.

•	It is better to provide more recent references for the readers to learn about this area, such as:

Zhao-Rong Lai, Haisheng Yang, Linear Trading Position with Sparse Spectrum, IJCAI 2025.

---

> ### Author Rebuttal · Authors · 2026-03-30
>
> We sincerely thank the reviewer for the careful reading and constructive comments. We are particularly encouraged that the reviewer recognizes the theoretical contribution of our work. We also appreciate the positive feedback on the problem motivation and the use of multiple financial evaluation metrics. We address the concerns point by point below.
> 1. On the lack of wall-clock time comparison.
> We agree that running time is an important practical aspect. In the revision, we will provide detailed running time results. For completeness, we already report the additional running time data in Table 1 below. It can be observed from the table that the proposed STCO methods have running times that mostly lie in the range of $10^{−5}$ to $10^{−3}$ seconds for one period across different datasets and transaction costs rates, which are comparable to those of most baseline methods. Therefore, the results indicate that solving the proximal subproblem via ADMM does not introduce significant computational overhead in practice, which can be attributed to the linear convergence of the outer PSGA iterations.
> 2. On the practical selection of the penalty parameter $c$.
> Thank you for raising this important point. The condition $c>2\sqrt{2}L$ in Theorem 3.3 provides a sufficient theoretical guideline for choosing $c$. In practice, however, we observed that the proposed method is not highly sensitive to the exact value of $c$ as long as it lies within a reasonable range, so exhaustive tuning is unnecessary. As shown in Table 2, the cumulative wealth remains very stable across different values of $c$. For example, on NYSE(N) with γ=0.25%, the cumulative wealth of STCO1 remains around 233.43, while that of STCO2 remains around $6.59×10^5$; on TSE with γ=0%, the cumulative wealth of STCO1 remains around 1440.2, and that of STCO2 remains around 83.03. These results indicate that the portfolio performance is not highly sensitive to the choice of $c$ within a moderate range. Therefore, in our experiments, $c$ was chosen from such a reasonable range. We will clarify this in the revised manuscript.
> 3. On adding more recent references.
> Thank you for the helpful references. We will carefully check these works and cite the relevant ones in the revised version.
>
> Overall, we are grateful for the reviewer’s positive evaluation and constructive recommendations. We believe the suggested clarifications and additional empirical results will further improve the quality and impact of the paper.
>
> **Table 1: The running times (in seconds) of different portfolio optimization models for one period on 4 data sets under transaction cost rates γ = 0%, 0.25%, and 0.5%**
>
> |Algorithm|NYSE(O) 0%|NYSE(O) 0.25%|NYSE(O) 0.5%|NYSE(N) 0%|NYSE(N) 0.25%|NYSE(N) 0.5%|TSE 0%|TSE 0.25%|TSE 0.5%|MSCI 0%|MSCI 0.25%|MSCI 0.5%|
> |-|-|-|-|-|-|-|-|-|-|-|-|-|
> |TCO1|1.82E-02|1.15E-04|3.96E-04|1.69E-05|7.13E-05|1.42E-05|1.90E-03|2.62E-05|1.31E-05|3.85E-05|9.80E-06|7.70E-06|
> |TCO2|8.27E-04|1.45E-04|4.30E-04|1.05E-05|1.35E-05|9.50E-06|2.72E-04|2.68E-05|1.05E-04|2.20E-05|7.80E-06|9.45E-05|
> |S1|1.51E-03|4.99E-05|8.38E-05|2.08E-04|4.60E-06|3.20E-06|3.53E-05|7.80E-06|3.60E-06|3.05E-05|3.30E-06|3.80E-06|
> |S2|6.63E-05|2.87E-05|3.22E-05|3.00E-06|1.40E-06|5.00E-07|4.50E-06|1.50E-06|1.20E-06|2.10E-06|1.10E-06|8.00E-07|
> |S3|2.81E-04|2.83E-05|2.52E-05|2.49E-04|2.21E-05|1.10E-06|4.10E-06|1.10E-06|1.30E-06|2.30E-06|1.00E-06|1.10E-06|
> |SSPO|1.16E-02|1.21E-02|1.07E-02|9.93E-03|8.44E-03|8.24E-03|3.44E-02|2.87E-02|2.80E-02|9.57E-03|9.93E-03|1.03E-02|
> |DENRPO1-PARM|8.60E-03|3.17E-03|2.71E-03|1.55E-03|9.68E-04|9.51E-05|9.80E-03|7.94E-03|3.82E-03|2.49E-02|6.94E-05|6.10E-05|
> |DENRPO1-OLMAR|4.95E-03|5.69E-03|8.43E-05|2.41E-03|1.58E-04|5.01E-05|2.09E-02|5.59E-03|1.35E-03|1.26E-02|6.07E-05|8.06E-05|
> |DENRPO2-PARM|1.88E-03|2.98E-04|1.12E-03|2.28E-04|8.09E-04|2.87E-05|4.97E-04|7.18E-04|3.40E-03|3.63E-04|2.68E-05|3.12E-05|
> |DENRPO2-OLMAR|1.95E-03|1.01E-03|5.98E-05|2.65E-04|7.82E-05|1.84E-05|1.37E-04|3.45E-03|4.64E-05|5.93E-04|2.10E-05|2.13E-05|
> |STCO1|8.16E-03|2.20E-03|2.48E-03|8.87E-05|4.38E-05|2.50E-05|2.40E-02|1.43E-02|1.68E-02|8.64E-05|2.93E-05|2.96E-05|
> |STCO2|1.15E-03|9.41E-05|8.57E-05|5.86E-04|8.64E-05|1.11E-05|6.71E-03|1.70E-03|5.41E-04|5.38E-05|1.27E-05|1.30E-05|
>
> **Table 2: Cumulative wealth of STCO1 and STCO2 under different values of c on the NYSE(N) dataset with γ = 0.25% and on the TSE dataset with γ = 0%**
>
> |c values|NYSE(N) (γ=0.25%)|NYSE(N) (γ=0.25%)|TSE (γ=0%)|TSE (γ=0%)|
> |-|-|-|-|-|
> ||STCO1|STCO2|STCO1|STCO2|
> |2E-6|233.43|6.59E+5|1440.27|83.02|
> |4E-6|233.43|6.59E+5|1440.28|83.03|
> |8E-6|233.43|6.59E+5|1440.28|83.03|
> |1.6E-5|233.42|6.59E+5|1440.16|83.03|

---

> > ### Author Rebuttal · Reviewer_rJQ5 · 2026-04-03
> >
> > The authors have fully addressed my questions, thus I have raised the score to recommend an acceptance of this paper.

---

> > > ### Author Response · Authors · 2026-04-07
> > >
> > > Thank you very much for your positive feedback and for raising your score to recommend acceptance. We are glad that our responses have fully addressed your concerns. We sincerely appreciate your thoughtful comments, which have helped improve both the clarity and quality of our work.

---

### Official Review · Reviewer_NRVc · 2026-02-27

**Soundness:** 3
**Presentation:** 3
**Significance:** 2
**Originality:** 2
**Overall Recommendation:** 4
**Confidence:** 4

**Summary:**

This paper proposes an optimization approach for sparse portfolio optimization with transaction costs. The authors consider the problem of short-term portfolio optimization under proportional transaction costs and a constraint limiting the number of assets included in the portfolio. They recast the problem as a difference of convex optimization problem by reformulating the cardinality constraint and use a proximal subgradient algorithm to find local optima.

**Compliance With Llm Reviewing Policy:**

Affirmed.

**Final Justification:**

The rebuttal addressed all my concerns regarding presentation issues, which changed my overall recommendation.

**Key Questions For Authors:**

- Why has the proximal operator no closed-form solution, as claimed by the authors in section 3.3?

**Limitations:**

The authors should discuss in more depth the societal implications of their work, since it involves optimizing investment portfolio which can have huge economic consequences and societal implications.

**Strengths And Weaknesses:**

- Soundness: The reformulation of the optimization problem is sound and the PSGA algorithm seems correct. However, I believe their formulation of transaction costs in the first equation of 2.2 (equation unnumbered) to be wrong, since $\hat{\mathbf{w}}_{[t-1]}$ represents relative allocation and the second term in the $\lVert \cdot \rVert_1$ part is $\mathbf{w}_t$$\theta$  which is a wealth term. However, the formulation in eq. (4) is correct, so this seems a minor error. Additionally, the definition of $f_c$ before Theorem 3.3 is not consistent with eq. (8).
- Presentation: The paper has several presentation issues, the most pressing of which is that the authors reference to main results in the text that are located in the appendix. I refer specifically to Table 2 in Appendix A.16 which obviously is part of the main results, since the authors include it in their main discussion and cannot be considered as supplementary material. Additionally, the presentation of the some theoretical results is unclear, for instance in section 3.3 the authors state that the proximity operator has no closed-form solution without justifying it. Finally, the authors citation style in many cases with no year and only author name does not comply with the citation conventions, for instance Wang et. al. in the introduction.
- Significance: The problem of short-term portfolio optimization with transaction costs is well known in the literature. While the method presented is new, it only guarantees finding local optima, which might be very far from the global optimum. The authors should argue why their method is better than other state-of-the-art heuristics. The authors only compare cumulative wealth in the main text via Figure 1, which is insufficient to demonstrate the practical significance of the method.
- The PSGA is novel, however the method itself reduces an optimization problem to an optimization problem with three embedded additional optimization problems to estimate the proximity operator. While the authors present a novel approach, it is unclear how this advances the field of improves the understanding of the problem.

---

> ### Author Rebuttal · Authors · 2026-03-30
>
> We thank the reviewer for the detailed comments. We respectfully clarify that several concerns arise from misunderstandings of the model formulation and the scope of our claims.
> 1. On the transaction cost modeling.
> We do not agree that the transaction cost formulation is incorrect. The expression in Section 2.2 follows the standard modeling in (Li et al., 2017), where $ŵ_{[t-1]}$ denotes the allocation at the end of period $t−1$, and $w_{[t]}\theta_{[t-1]}$ is the starting wealth for period $t$. Therefore, the  $||ŵ_{[t-1]}-w_{[t]}\theta_{[t-1]}||_1$ captures the difference of wealth during rebalancing, and hence correctly models transaction costs rate. We will revise the manuscript to make this interpretation clearer.
> 2. On the consistency between $f_c$ and equation (8).
> There is no inconsistency. Equation (8) is equivalent to minimizing a term of the form $c(‖w‖₁-‖w‖_{(K)})$.  As pointed out in the paper, the simplex constraint implies $||w||_1=1$, which is a constant that can be omitted in the objective. Thus, the definition of $f_c$ before Theorem 3.3 is fully consistent with equation (8).
> 3. On the presentation.
> We agree that the presentation can be improved, but these issues do not affect the soundness or significance of the work. First, Table 2 was placed in the appendix solely because of the ICML page limit and its large size, we will move a summarized version of Table 2 into the main text and retain the complete table in the appendix. Besides, we will also revise the citation style to fully comply with ICML conventions.
> 4. On the significance of problem setting.
> We respectfully disagree that the problem is already well known and that the contribution is therefore limited. While short-term portfolio optimization with transaction costs has been studied, simultaneously incorporating transaction costs, variable sparsity, and short-term portfolio setting is rare in existing literature. Most prior works either ignore sparsity or focus on long-term or mean-variance settings. This is exactly the gap our work targets.
> 5. On “only local optima”.
> We do not believe this criticism is fair. The original model contains an NP-hard $\ell_0$ constraint. For such problems, pursuing a globally optimal algorithm would usually entail prohibitive computational cost and is therefore often less attractive in practice. The paper proves that the proposed DC reformulation and the original $\ell_0$ problem share the same global (local) optimal solutions in mild conditions. Moreover, full-sequence convergence to a critical point at a linear rate is a strong result for a nonsmooth nonconvex cardinality problem.
> 6. On superiority over heuristics.
> Our method is preferable for two reasons. First, theoretical guarantees. We establish exact penalty equivalence, full-sequence convergence, KL analysis, and R-linear convergence. This is substantially stronger than the guarantees available for many heuristic or learning-based competitors. Second, interpretability. Unlike black-box heuristics, the model explicitly encodes both sparsity and transaction cost, making the portfolio directly understood.
> 7. On empirical evidence.
> We disagree that Figure 1 is the only evidence of practical significance. In addition to cumulative wealth, the paper reports MER, alpha-factor, and beta-factor, which jointly evaluate return and risk. These are standard metrics in portfolio evaluation, and the results show that our method achieves higher returns with lower risk than state-of-the-art algorithms in most cases. In the revision, we will further include comparisons across different sparsity levels and additional running time results to better demonstrate the practical advantages of the proposed method. Due to the rebuttal length limit, detailed additional experiments results are provided in our responses to the other reviewers, or available at
> https://anonymous.4open.science/r/STCO-Experimental-Results-900F.
> 8. On the algorithmic contribution.
> We disagree with the claim that the method does not advance the field. The contributions are not merely procedural for the specific problem, but rather: (i) an exact DC reformulation of an NP-hard problem with theoretical guarantees; (ii) a proximal subgradient framework adapted to the nonsmooth DC structure; (iii) a new auxiliary function enabling global convergence of the full sequence and an R-linear convergence rate via a KL exponent of 1/2. This is a coherent methodological and theoretical contribution.
> 9. On the key question of proximal operator.
> The absence of a closed form is structural. The proximal operator in (11) leads to the subproblem combining a quadratic term, a shifted nonsmooth $\ell_1$ term, and a simplex constraint. Unlike classical cases (e.g., pure $\ell_1$  or simplex projection), this combined structure does not admit a separable or closed-form solution. This is precisely why we employ ADMM to compute it in Section 3.3. Thus, the absence of a closed form is not an unsupported claim.

---

> > ### Author Rebuttal · Reviewer_NRVc · 2026-04-05
> >
> > I thank the reviewers for their clarifications. While I still do not agree on some technical points, this cannot be addressed in a short rebuttal. As the presentation issues were resolved, and these were my major objections, I will raise my recommendation.

---

> > > ### Author Response · Authors · 2026-04-07
> > >
> > > Thank you for your thoughtful follow-up and for your positive reassessment of our paper. We are glad that the major objections have been addressed and that our clarifications were helpful. Regarding the remaining technical differences, we agree that a full discussion would require more space than allowed in the rebuttal. In the revised version, we will further clarify the related arguments to present them more clearly. We sincerely appreciate your careful reading and constructive feedback, which have helped improve the clarity of the paper.

---

### Official Review · Reviewer_JBBv · 2026-03-11

**Soundness:** 4
**Presentation:** 4
**Significance:** 4
**Originality:** 4
**Overall Recommendation:** 5
**Confidence:** 4

**Summary:**

This paper studies sparse portfolio optimization with transaction cost. The authors formulate a $K$-sparse transaction cost optimization (STCO) model that jointly considers expected return, $\ell_1$ transaction cost, and a cardinality constraint. To handle the combinatorial difficulty of the $\ell_0$ constraint, the model is reformulated into an equivalent difference-of-convex (DC) optimization problem using the top-$K$ norm representation. Based on this reformulation, the paper proposes a proximal subgradient algorithm (PSGA) to solve the problem. The authors further provide theoretical analysis showing that the generated sequence globally converges to a critical point, and under certain conditions the limit point corresponds to a local optimum of both the DC reformulated problem and the original  model. Moreover, the algorithm is shown to achieve $R$-linear convergence.

**Compliance With Llm Reviewing Policy:**

Affirmed.

**Final Justification:**

The authors have addressed my questions well, and I therefore raise my recommendation for acceptance.

**Key Questions For Authors:**

1. The experiments rely on the $K$-sparse constraint $ \|w\| _0 \le K $, but the paper does not clearly specify the value of $K$ used in the experiments. It would be helpful if the authors could clarify how the sparsity level $K$ is selected and whether the results are sensitive to this parameter.

2. Since each PSGA iteration requires solving a proximal subproblem via ADMM, could the authors provide more detailed runtime analysis to illustrate the practical computational cost?

3. The authors may consider citing several recent works related to sparse optimization and subgradient-based algorithms, for example:

   - Lai, Z.-R., & Yang, H. (2025). *Linear trading position with sparse spectrum*. In *Proceedings of the Thirty-Fourth International Joint Conference on Artificial Intelligence (IJCAI)*, pp. 5554–5562.

   - Lai, Z.-R., Wu, X., Fang, L., Chen, Z., & Li, C. (2025). *De-singularity Subgradient for the q-th-Powered $\ell_p$-Norm Weber Location Problem*. *Proceedings of the AAAI Conference on Artificial Intelligence*, 39(17), 18026–18034.

   - Lai, Z.-R., Wu, X., Fang, L., & Chen, Z. (2024). *A de-singularity subgradient approach for the extended Weber location problem*. In *Proceedings of the Thirty-Third International Joint Conference on Artificial Intelligence (IJCAI)*, pp. 4370–4379.

**Limitations:**

No. The discussion could be strengthened by elaborating on practical limitations such as parameter sensitivity.

**Strengths And Weaknesses:**

## Strengths

1. The proposed PSGA method uses the DC reformulation and combines proximal subgradient updates with ADMM to compute the proximal operator efficiently. The algorithmic structure is clear and implementable.

2. The paper provides convergence analysis for the proposed method. In particular, the authors show that the generated sequence globally converges to a critical point and, under certain conditions, the limit point corresponds to a local optimum of both the DC reformulated problem and the original model. Establishing convergence to a local optimum for a nonconvex problem is a nontrivial theoretical result. In addition, the authors establish $R$-linear convergence under the KL framework.


## Weaknesses

The experiments focus mainly on cumulative wealth and several financial metrics. Additional analysis (e.g., sensitivity to sparsity level $K$, or convergence behavior of PSGA) could strengthen the empirical evidence.

---

> ### Author Rebuttal · Authors · 2026-03-30
>
> We thank the reviewer for the positive feedback and constructive suggestions. We appreciate the recognition of the PSGA-ADMM framework and the theoretical contribution. In the following, we address the points raised.
> 1. On the choice of the sparsity level $K$.
> Thank you for pointing out the lack of clarity in the description of $K$. In our experiments, we set $K=10$, but this was not clearly stated,  and we will correct this in the revision. To clarify the effect of $K$, we conducted additional experiments for $K=10$ and $K=15$ (Table 1). The cumulative wealth results remain almost unchanged, showing that the method is robust to K within a moderate range and that using $K=10$ is a reasonable choice. Furthermore, Table 2 shows the average and standard deviation of sparsity of the portfolios generated by STCO1 and STCO2 and other state-of-the-art algorithms with $K=10$. We observe that the average sparsity of STCO1 is about 1.0531 on NYSEO when γ=0%, and similar values can also be observed across other datasets and settings. This indicates that the proposed STCO method selects only a small number of assets while maintaining strong portfolio performance.
> 2. On the runtime cost of solving the proximal subproblem via ADMM.
> We agree that running time is an important practical aspect. In the revision, we will provide detailed running time results. Due to the rebuttal character limit, the additional running time data can be referenced in Table 1 of Reviewer rJQ5's response, or full experimental results are available at: https://anonymous.4open.science/r/STCO-Experimental-Results-900F. We observe from the table that the proposed STCO methods have running times that mostly lie in the range of $10^{−5}$ to $10^{−3}$ seconds for one period across different datasets and transaction costs rates, which are comparable to those of most baseline methods. Therefore, the results indicate that solving the proximal subproblem via ADMM does not introduce significant computational overhead in practice, which can be attributed to the linear convergence of the outer PSGA iterations.
> 3. On the suggested related works. Thank you for the helpful references. We will carefully check these works and cite the relevant ones in the revised version.
>
> Overall, we are grateful for the reviewer’s positive evaluation and constructive recommendations. We believe the suggested clarifications and additional empirical results will further improve the quality and impact of the paper.
>
> **Table 1.Cumulative wealth of STCO1 and STCO2 under sparsity levels K=10 and K=15 in TSE and MSCI datasets.**
>
> |algorithms\(γ = 0\%\)|TSE|TSE|MSCI|MSCI|
> |-|-|-|-|-|
> ||K = 10|K = 15|K = 10|K = 15|
> |STCO1|1440.28|1440.28|10.19|10.19|
> |STCO2|83.03|83.03|13.99|13.99|
>
> |algorithms\(γ = 0.25\%\)|TSE|TSE|MSCI|MSCI|
> |-|-|-|-|-|
> ||K = 10|K = 15|K = 10|K = 15|
> |STCO1|6.97|6.97|2.84|2.84|
> |STCO2|3.94|3.94|1.98|1.98|
>
> **Table 2.Average and standard deviation of sparsity of the portfolios generated by STCO1 and STCO2 and other state-of-the-art algorithms under transaction cost rate γ= 0% and 0.25%**
>
> |Algorithm|Statistic|NYSE(O)|NYSE(O)|NYSE(N)|NYSE(N)|TSE|TSE|MSCI|MSCI|
> |-|-|-|-|-|-|-|-|-|-|
> |||γ=0%|γ=0.25%|γ=0%|γ=0.25%|γ=0%|γ=0.25%|γ=0%|γ=0.25%|
> |S1|mean|1.0488|1.0488|1.0706|1.0706|1.0921|1.0921|1.2061|1.2061|
> |S1|std|0.5119|0.5119|0.6573|0.6573|2.4562|2.4562|1.4651|1.4651|
> |S2|mean|1.0062|1.0062|1.0034|1.0034|1.0691|1.0691|1.0221|1.0221|
> |S2|std|0.4656|0.4656|0.2743|0.2743|2.4519|2.4519|0.7122|0.7122|
> |S3|mean|1.0487|1.0487|1.0580|1.0580|1.0921|1.0921|1.1438|1.1438|
> |S3|std|0.5110|0.5110|0.3664|0.3664|2.4562|2.4562|0.8045|0.8045|
> |TCO1|mean|7.6556|7.5861|7.5116|7.4886|6.7665|6.7482|10.4688|10.6616|
> |TCO1|std|3.5488|3.5078|3.2425|3.2964|4.1446|4.1448|3.8044|4.1774|
> |TCO2|mean|6.2674|6.1996|5.9793|5.9364|5.5941|5.5909|8.2042|8.1668|
> |TCO2|std|3.0564|3.0300|2.9270|2.9246|3.9788|3.9549|3.4682|3.7179|
> |SSPO|mean|1.0143|1.0143|1.0089|1.0089|1.0778|1.0778|1.0297|1.0297|
> |SSPO|std|0.4741|0.4741|0.2840|0.2840|2.4534|2.4534|0.7173|0.7173|
> |DENRPO1-PARM|mean|1.0322|1.0448|1.0361|1.1406|1.0818|1.1160|1.2550|2.5216|
> |DENRPO1-PARM|std|0.5052|0.5152|0.3341|0.6991|2.4541|2.4940|2.0577|3.9176|
> |DENRPO1-OLMAR|mean|1.0379|1.0848|1.0373|1.1246|1.0921|1.2836|1.0738|5.7948|
> |DENRPO1-OLMAR|std|0.4978|0.8616|0.3316|1.0319|2.4562|3.9862|0.7469|5.8775|
> |DENRPO2-PARM|mean|1.0322|1.0444|1.0361|1.1306|1.0818|1.1160|1.2550|2.8217|
> |DENRPO2-PARM|std|0.5052|0.5149|0.3341|0.6618|2.4541|2.4940|2.0577|4.2794|
> |DENRPO2-OLMAR|mean|1.0379|1.0846|1.0373|1.1235|1.0921|1.2828|1.0738|5.7948|
> |DENRPO2-OLMAR|std|0.4978|0.8615|0.3316|1.0128|2.4562|3.9862|0.7469|5.8775|
> |STCO1|mean|1.0531|1.0821|1.0620|1.1827|1.0961|1.1461|1.3126|2.8849|
> |STCO1|std|0.5248|0.5546|0.3721|0.7330|2.4565|2.4985|2.0675|4.2975|
> |STCO2|mean|1.0541|1.1308|1.0571|1.1549|1.1064|1.3193|1.1189|2.1381|
> |STCO2|std|0.5145|0.8857|0.3602|0.8471|2.4582|4.0208|0.7783|3.9186|

---

> > ### Author Rebuttal · Reviewer_JBBv · 2026-04-03
> >
> > The authors have addressed my questions well, and I therefore raise my recommendation for acceptance.

---

> > > ### Author Response · Authors · 2026-04-07
> > >
> > > Thank you very much for your positive feedback and for raising your recommendation for acceptance. We are glad that our responses have adequately addressed your concerns. We sincerely appreciate your careful reading and constructive comments, which have helped improve the quality and clarity of our work.

---

### Decision · Program_Chairs · 2026-04-30

**Decision:**

Accept (regular)

**Comment:**

This paper proposes the STCO model for sparse portfolio optimization with transaction costs, along with a proximal subgradient algorithm equipped with ADMM for efficient computation. It establishes global convergence and R-linear convergence guarantees via the KL property. The submission received three detailed, broadly positive reviews: two reviewers endorse acceptance, highlighting its novel DC reformulation, rigorous theoretical analysis, and comprehensive empirical evaluation. The third reviewer recommends weak acceptance, noting minor concerns regarding presentation, local optimality, and empirical depth. All issues were thoroughly addressed in the author responses with clear clarifications and additional experimental results.

Based on the reviews and author responses, this work offers novel and solid contributions to the field of sparse portfolio optimization. The meta reviewer accordingly recommends the paper for acceptance. The authors are strongly encouraged to implement all revisions and clarifications promised in their rebuttal when preparing the final manuscript.